# Circumferential actomyosin bundles anchored by CCM1 drive endothelial cell contraction and vessel constriction

Yan Chen[1], Nuria Taberner[1], Jason da Silva[1,2], Vivek Semwal[1], Biplab Bhattacherjee [1], Julia Eckert [2], Igor Kondrychyn [1], Mingzhao Hu[1,3], Nitish Aswani[1], Guihua Chen[1], Yasushi Okada[1,4,5,6,7], Anne Karine Lagendijk [2,8], Tatsuo Shibata [1], Satoru Okuda [9] & Li-Kun Phng [1] ✉

Blood vessels undergo extensive remodelling to acquire appropriate diameters, yet how endothelial cells coordinate changes in their number and shape to achieve this remains unclear. Here we show that endothelial cell contraction and rearrangements underlie the inverse relationship between cell number and vessel diameter during development. Using high-resolution imaging and manipulation of actin cytoskeleton organisation, in vivo laser ablation experiments and mathematical simulations, we reveal that tension-bearing, circumferential actomyosin bundles form in the endothelial cortex to drive endothelial cell contraction and vessel constriction. The anchorage of circumferential actin bundles to cell-cell junctions is mediated by Ccm1/Krit1 protein. Importantly, the loss of circumferential actin bundles in *ccm1*-deficient endothelial cells causes cell enlargement and impaired vessel constriction, culminating in vessel dilation characteristic of cerebral cavernous malformations. Our multiscale study demonstrates how circumferential actomyosin-driven endothelial cell contractions regulate vessel diameter and provides insights into mechanisms of both normal vascular development and disease pathogenesis.

The tight control of blood vessel diameter is crucial for the efficient distribution of blood flow and tissue perfusion. During development, vessel networks are refined through vascular remodelling, a process that integrates elongation, pruning, and diameter regulation to establish a hierarchical network of arteries, veins, and capillaries[1,2]. The mis-regulation of vessel size can culminate in vascular malformations such as Hereditary Haemorrhagic Telangiectasia (HHT), which is characterised by enlarged arteries and veins that connect with each other to create shunts, and Cerebral Cavernous Malformation (CCM), which presents as clusters of abnormally dilated blood vessels in the brain[3–5]. In many vascular malformations, blood flow distribution through the vascular network is impaired and can result in bleeding and chronic pain[6]. Understanding how vessel diameter is defined at the cellular and molecular levels is therefore fundamental to vascular biology and disease.

[1]RIKEN Center for Biosystems Dynamics Research, Kobe, Japan. [2]Institute for Molecular Bioscience, The University of Queensland, Brisbane, Queensland, Australia. [3]Department of Biological Sciences, Graduate School of Science, The University of Osaka, Osaka, Japan. [4]Department of Cell Biology, Graduate School of Medicine, The University of Tokyo, Tokyo, Japan. [5]Department of Physics, Graduate School of Science, The University of Tokyo, Tokyo, Japan. [6]Universal Biology Institute (UBI), The University of Tokyo, Tokyo, Japan. [7]International Research Center for Neurointelligence (WPI-IRCN), Institutes for Advanced Study, The University of Tokyo, Tokyo, Japan. [8]School of Biomedical Sciences, Faculty of Medicine, The University of Queensland, Brisbane, Queensland, Australia. [9]Nano Life Science Institute, Kanazawa University, Kanazawa, Japan. ✉e-mail: likun.phng@riken.jp

Endothelial cells (ECs) are the primary building blocks of vessels, and their number and size have been recognised as critical determinants of vessel diameter. Increased EC numbers through vessel fusion[7], directed migration[8–10], or excess proliferation can enlarge vessels[1]. In addition to changes in cell number, alterations in individual EC size represent another key mechanism for remodelling vessel diameter[11,12]. PlexinD1-KLF2[13], Bone Morphogenetic Protein (BMP) and transforming growth factor beta (TGFb) signalling regulate multiple aspects of EC behaviour, including cell size. Disruption of TGFb signalling, including mutations in Alk1 and its coreceptor Endoglin[11,14], or the downstream mediator, SMAD4[15], results in enlarged ECs that lead to aberrations in vessel diameter as observed in HHT. Similarly, EC enlargement occurs in CCM models with KRIT1/CCM1 mutation[16–18]. While these findings highlight EC size control as an important factor in the regulation of vascular morphology, the underlying biophysical mechanisms remain poorly understood. As recent studies have implicated the role of actin cytoskeletal remodelling in controlling EC size and vessel diameter[12,19,20], we speculate that alterations in EC mechanics may underlie the pathogenesis of vascular malformations.

Here, using zebrafish intersegmental vessels (ISVs) as a model, we investigated the self-organising principles of blood vessel remodelling across subcellular, cellular and tissue scales. ISVs undergo the seemingly paradoxical behaviour of radial constriction despite an overall increase in EC number from 2 to 4 days post fertilisation (dpf). By combining live imaging with quantitative strain analysis, we resolved this paradox: vessel elongation is accommodated by additional ECs, while vessel constriction is achieved through active EC contraction and rearrangement. At the subcellular level, we identify dynamic cortical actin organisations and demonstrate that circumferential actin bundles, reinforced by non-muscle myosin II and anchored by Krit1 at junction-cortex interfaces, generate tensile forces that progressively contract ECs to narrow vessel diameter. When this cytoskeletal architecture is disrupted, either by altering cortical actin organisation, inhibiting myosin II or loss of Krit1, ECs fail to contract, rearrange abnormally, and vessels dilate. Conversely, the overexpression of Krit1 enhances circumferential actin assembly and vessel constriction, underscoring its function as a key organiser of the cortical contractile machinery. This work therefore not only provides mechanistic insight into how actin cytoskeleton organisation controls vessel diameter during normal blood vessel development but also explains how its disruption leads to pathological dilation in CCM. Overall, our study establishes a framework for understanding how EC mechanics integrate with cell number and genetic regulators to generate vessels of appropriate size.

## Results

### Dynamic endothelial cell behaviours underly vessel constriction and elongation

To investigate temporal changes in vessel morphometrics during vascular remodelling, we examined zebrafish arterial and venous intersegmental vessels (aISVs and vISVs) from 2 to 4 dpf (Fig. 1a). aISVs displayed a gradual, significant reduction in diameter over this period, whereas vISVs showed similar diameter between 2-3 dpf and constricted between 3-4 dpf (Fig. 1b). Quantification of EC nuclei revealed a substantial increase in cell number within both aISVs and vISVs (Fig. 1c). Although EC number has been reported to drive vessel size enlargement[7,8], we observed an inverse relationship between increasing cell number and vessel diameter. Vessel length increased (Fig. 1d), and the average EC number per unit length (100 μm) showed a similar upward trend in both vessel types (Supplementary Fig. 1a), indicating a positive correlation between EC number and vessel elongation. Thus, elongation accommodates additional cells, while circumferential space becomes limited as diameter decreases. Consistently, average vessel area changed only modestly (Supplementary Fig. 1b), suggesting that ECs must deform to fit within the remodelling vessel.

To elucidate the source of EC number increase, we performed time-lapse imaging in *Tg(fli1:Lifeact-mCherry)[ncv7]; Tg(fli1a:H2B-EGFP)[ncv69]* embryos from 2 to 4 dpf. In aISVs, most (>50%) underwent only positional rearrangements within the vessel. From 2-4 dpf, ~30% of vessels exhibited EC exchange with the dorsal longitudinal anastomotic vessel (DLAV) while ~10% displayed both exchange and cell division (Fig. 1e), with most exchange events involving DLAV-to-aISV immigration (Supplementary Fig. 1c, d). Thus, aISV cell number arises primarily from DLAV-derived cell immigration. In vISVs, ~80% of vessels displayed both EC division and exchange between 2-3 dpf, and ~30% still showed both behaviours between 3-4 dpf (Fig. 1f). Cell exchanges mainly involved vISV-to-DLAV emigration, with minimal contribution from the PCV (Supplementary Fig. 1e, f). Nearly all ECs of vISVs underwent mitosis, with most dividing two or more times, indicating that cell division is the main driver of venous EC number increase (Supplementary Fig. 1g). Thus, vISVs grows mainly through cell division while also serving as a cell source for the DLAV, whereas aISVs grows primarily through DLAV-derived exchange (Fig. 1g).

### Endothelial cell shrinkage and rearrangement drive vessel constriction and elongation

To assess whether EC deformation contributes to ISV remodelling, we unwrapped individual ECs using a custom ImageJ script and quantified their 2-dimensional (2D) area and aspect ratio (Fig. 2a). EC significantly reduced in size in both aISVs and vISVs between 2 and 4 dpf (Fig. 2b), with similar cell aspect ratio (Supplementary Fig. 1h), suggesting that ECs shrink in both radial and axial directions. Because vessels also undergo changes in both diameter and length, we next assessed cell behaviours along the radial (circumferential) and axial (longitudinal) axes separately.

To distinguish whether vessel constriction arises from ECs becoming narrower or from cells rearranging away from the vessel circumferential surface, we applied a strain analysis. Three parameters were calculated: (i) cell number strain, reflecting expected changes in how many cells are required to span the vessel axis independent of cell shape; (ii) cell strain (in aEC or vEC), reporting whether individual ECs shrink or expand; and (iii) vessel strain (in aISV or vISV), the net vessel-level change. Together, these measures allow us to distinguish whether remodelling is driven by cell deformation, cell rearrangement, or both.

In aISVs along the radial axis (Fig. 2c), positive cell number strain between 2 and 3 dpf reflected the addition of DLAV-derived cells. Despite this, vessel strain was negative because ECs narrowed (negative cell strain), counteracting circumferential widening due to increased cell number. By 3-4 dpf, negative cell number strain showed that fewer cells encircled the circumference as they rearranged axially. Combined with continued cell narrowing, this led to a clear negative vessel strain and net constriction. Along the axial axis (Fig. 2d), cell addition and axial rearrangement supported vessel elongation (positive cell number strain, positive vessel strain) despite cell shortening from 2 to 4 dpf. Thus, aISV constriction is driven by persistent EC shrinkage and reinforced by axial rearrangement.

In vISVs, high cell division between 2 and 3 dpf generated positive radial cell number strain, yet cell narrowing (negative cell strain) offset vessel widening, keeping diameter almost unchanged. By 3-4 dpf, axial rearrangement (negative cell number strain) rather than cell narrowing produced vessel constriction (Fig. 2e). Axial elongation was supported by persistent cell division and axial rearrangement (positive cell number strain) between 2-4 dpf, overcoming cell shortening (negative cell strain) (Fig. 2f). Thus, the mechanism of vISV constriction shifts from cell shrinkage at 2-3 dpf to cell rearrangement at 3-4 dpf (Fig. 2g).

### Time-lapse imaging reveals distinct cellular strategies for vessel constriction in aISVs and vISVs

To validate our static analyses with dynamic observations, we performed time-lapse imaging of ISVs from 2 to 3 dpf and 3 to 4 dpf. Cell

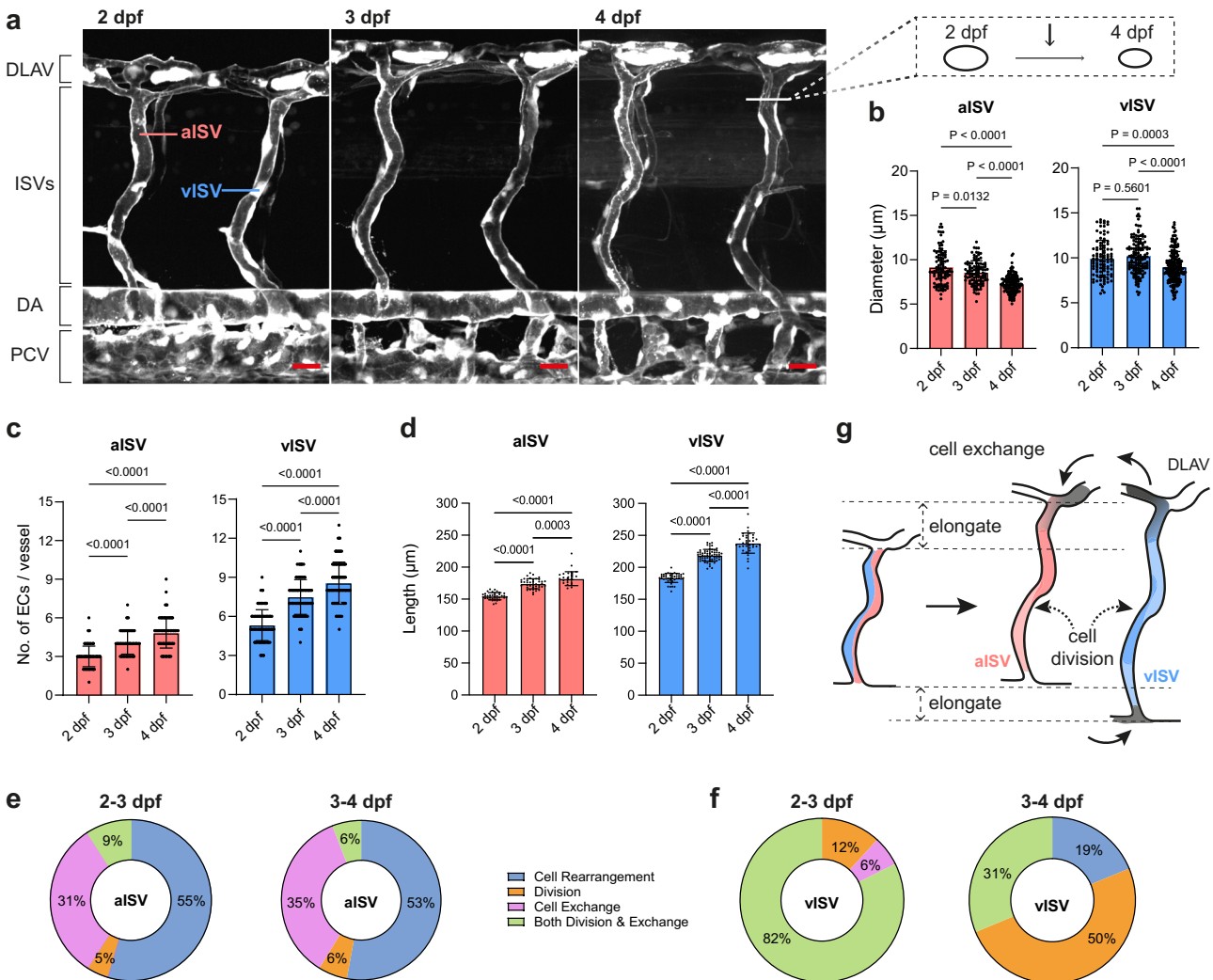

**Fig. 1 | Dynamic endothelial cell behaviours drive vessel constriction and elongation during ISV remodelling. a** Maximum intensity projection of confocal z-stacks of the same ISVs in zebrafish trunk from 2 to 4 dpf. *Tg(fli1:h2bc1-mCherry)^{ncv3l}* and *Tg(fli1:MYR-EGFP)^{ncv2}* label endothelial cell nuclei and cytosol, respectively; both channels are merged and displayed in greyscale. Scale bar, 20 μm. **b** Quantification of vessel diameter from 2–4 dpf in arterial ISVs (aISVs) and venous ISVs (vISVs) (aISV: n = 102/106/147 vessels; vISV: *n* = 97/138/189 vessels, from 21/30/33 embryos at 2/3/4 dpf, 3/6/6 experiments). **c** Quantification of EC number per vessel (aISV: *n* = 273/138/117 vessels; vISV: n = 266/177/134 vessels, from 31/31/27 embryos at 2/3/4 dpf, 3/2/2 experiments). **d** Quantification of vessel length from 2-4 dpf (aISV: *n* = 36/40/30 vessels; vISV: *n* = 42/60/34 vessels, from 20/25/16 embryos at 2/3/4 dpf, 2/2/1 experiments). Statistical significance in b-d was assessed by one-way ANOVA with Tukey's multiple comparisons test. Data are shown as Mean ± SD. **e, f** Percentage of ECs undergoing rearrangement, exchange, division, or combinations of these events in aISVs and vISVs (2-3 dpf: aISV, *n* = 22; vISV, *n* = 17; from 11 embryos and 2 experiments; 3-4 dpf: aISV, *n* = 17; vISV, *n* = 16; from 11 embryos and 2 experiments). **g** Schematic illustration of dynamic changes in vessel morphology and associated EC behaviours. DA dorsal aorta, DLAV dorsal longitudinal anastomotic vessel, PCV posterior cardinal vein, ISV intersegmental vessel, aISV arterial ISV, vISV venous ISV. Source data are provided as a Source Data file.

boundaries at the front and back sides (Supplementary Fig. 2a) of the vessel were visualised with an actin reporter, (*Tg(fli1:GAL4FF)^{ubs3}; Tg(UAS:EGFP-UCHD)^{ubs18}*, hereafter referred as EGFP-UCHD, enabling direct observation of dynamic cell movements.

In aISVs, cells frequently rearranged their positions while changing shape and size (Supplementary Movie 1). New cells entered from the DLAV and migrated ventrally against flow, displacing neighbouring cells as they occupied the vessel (Fig. 3a, bi, cell labelled in blue). During these rearrangements, a transient self-seam junction formed, in which a single EC (Fig. 3a, bii, cell labelled in pink) wrapped around the entire vessel circumference (59 and 63 hpf) to form a unicellular vessel segment before unwrapping to allow rearrangement of neighbouring cells (67 hpf). Coronal views showed that such cells underwent repeated inward-outward oscillations (Fig. 3a, bii). Self-seam junctions and the accompanying rearrangements were consistently observed in all examined aISVs

(*n* = 22/22). In addition, cells rearranged axially, allowing redistribution of cellular occupancy along the vessel circumference (Fig. 3a, bii, Supplementary Fig. 3a, b and Supplementary Movie 2). In contrast, vISVs remained multicellular, with frequent cell divisions (marked by the F-actin accumulation in rounded cells, Fig. 3c, Supplementary Fig. 3c and Supplementary Movies 3 and 4). At the same time, cells decreased in size and migrated dorsally toward the DLAV, often contributing to it. Thus, vISVs accommodated cell division through cell shrinkage and axial rearrangement.

Together, these analyses confirm that aISVs constrict through persistent cell shrinkage, combined with axial rearrangement and junction remodelling, whereas vISVs constrict under conditions of high cell division, where EC shrinkage and axial rearrangement accommodate additional cells. Interestingly, cortical actin stripes were occasionally observed (Fig. 3b, black arrows), prompting us to investigate cortical actin organisation in greater detail.

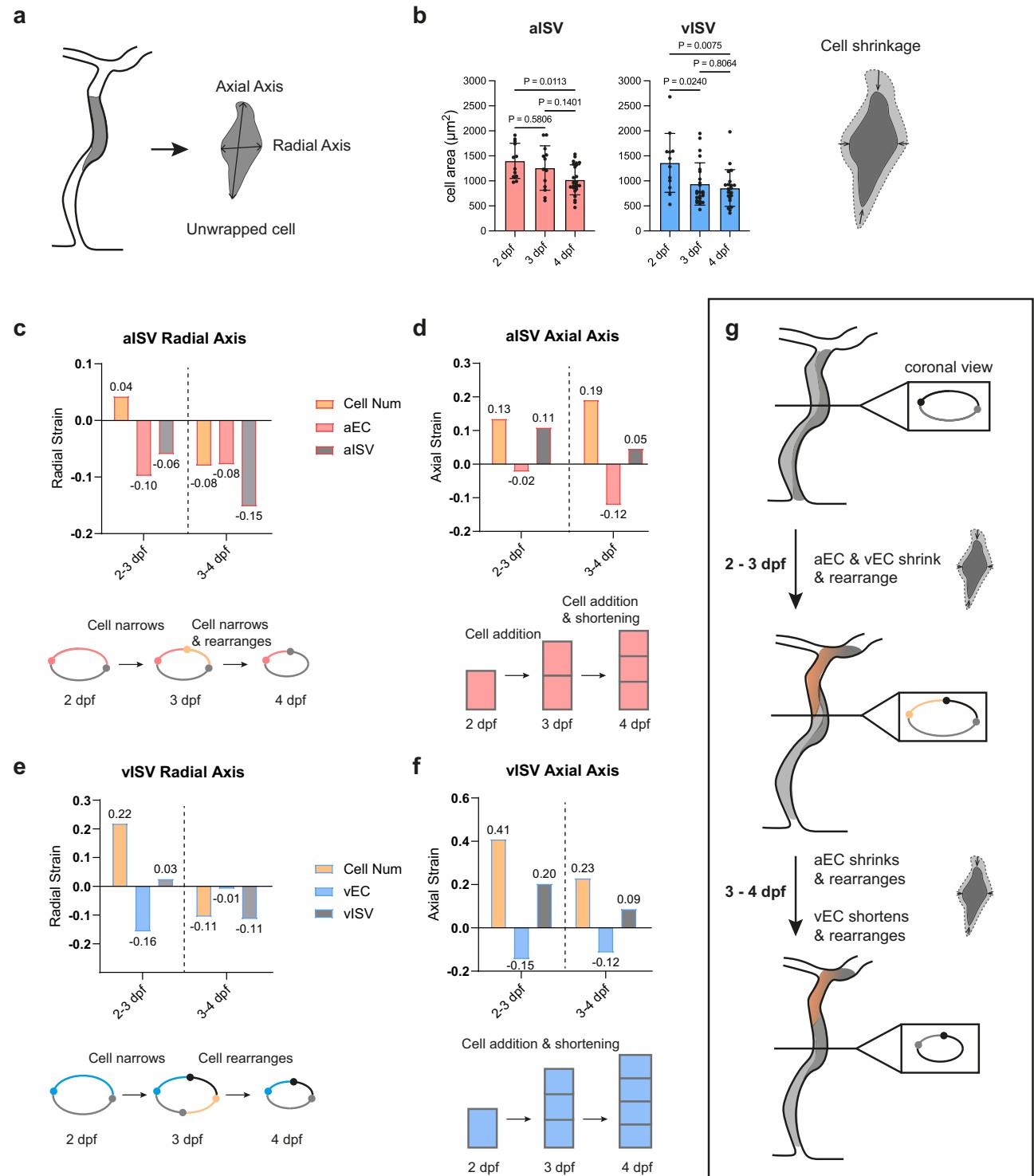

**Fig. 2 | Endothelial cell shrinkage and rearrangement underlie vessel constriction and elongation. a** Schematic illustration of radial and axial axes in an unwrapped EC. **b** Cell area in aECs and vECs from 2-4 dpf (aEC: $n = 12/13/24$ cells; vEC: $n = 12/25/21$ cells at 2/3/4 dpf; from 12/16/17 embryos at 2/3/4 dpf, 4 experiments). Statistical significance was assessed by one-way ANOVA with Tukey's multiple comparisons test. Data are shown as mean ± SD. **c**, **d** Strain analysis at radial (**c**) and axial (**d**) axes in aISVs between 2 and 4 dpf. Yellow bars, cell number strain; red bars, cell strain; grey bars, vessel strain. Values are indicated above/

below each bar. **e**, **f** Strain analysis at radial (**e**) and axial (**f**) axes in vISVs between 2 and 4 dpf. Yellow bars, cell number strain; blue bars, cell strain; grey bars, vessel strain. Values are indicated above/below each bar. **g** Schematics summarising aISV and vISV remodelling between 2 and 4 dpf, highlighting contributions of cell shrinkage and rearrangement. Shorten: cell length decreases at axial axis; Narrow: cell width decreases at radial axis; Shrink: reduces in both axial and radial axes. Source data are provided as a Source Data file.

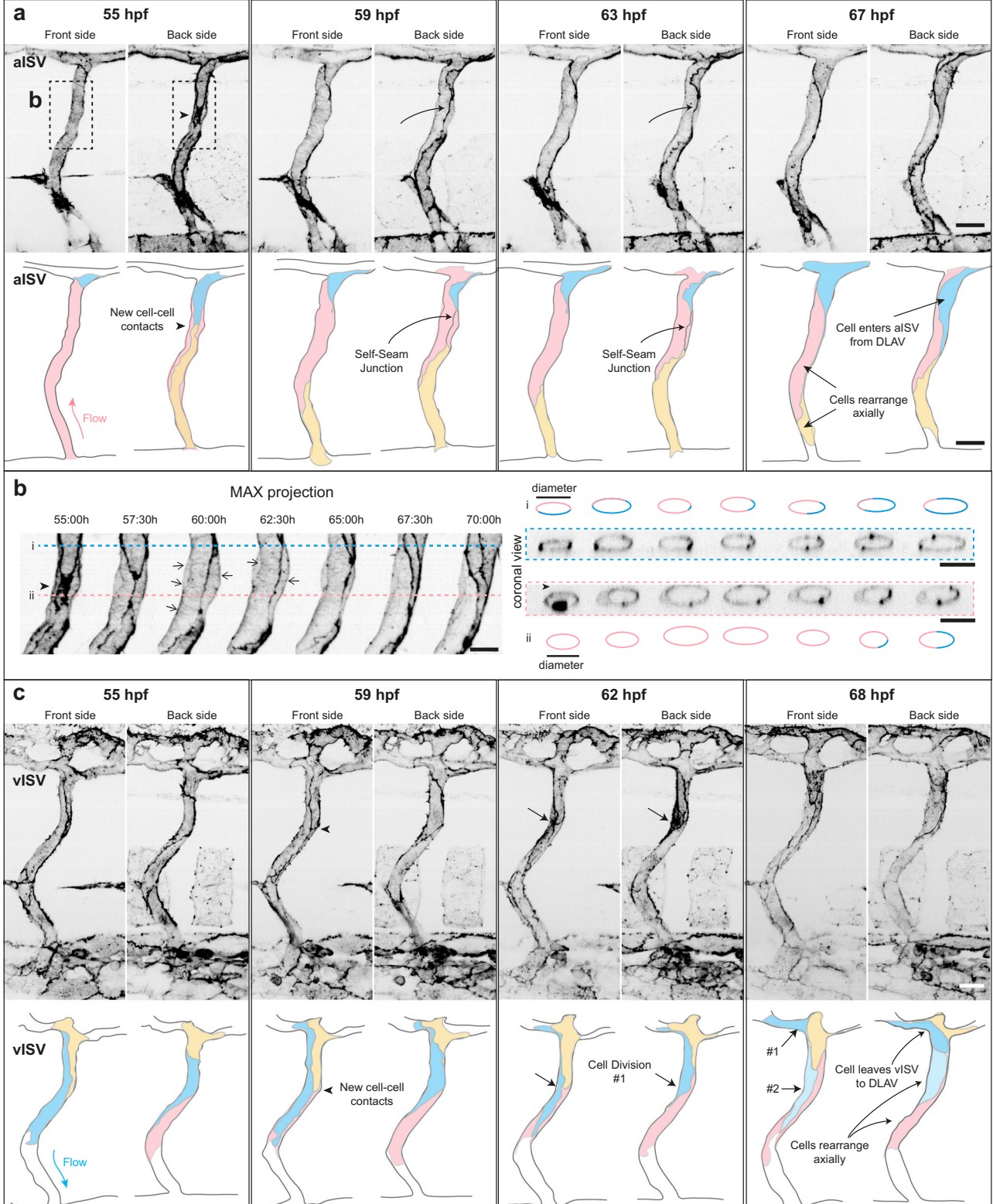

**Fig. 3 | Time-lapse imaging reveals distinct cell shape changes and rearrangement in aISVs and vISVs. a–c** Still images of an aISV (**a**, **b** Supplementary Movie 1) or a vISV (**c** Supplementary Movie 3) showing cortical and junctional actin in embryos from *Tg(fli1:GAL4FF)*[ubs3]; *Tg(UAS:EGFP-UCHD)*[ubs18] at 2-3 dpf. Vessels are shown as front (left) and back (right) sides. Schematics are traced from the original image and denote the constituent cells in colour (pink, blue, and yellow). Similar observations are made in 22 movies from 2 independent experiments. Scale bar, 20 μm. **b** Magnification of the inset in (**a**). Coronal views showed at z-plane indicated by blue serrated line (**i**) or a pink serrated line (**ii**). Scale bar, 10 μm. **c** Cell divisions are observed and indicated by black arrows. Daughter cells (#1 and #2) are denoted in dark and light blue. Similar observations are made in 17 movies from 2 independent experiments. Scale bar, 20 μm. ISV intersegmental vessel, aISV arterial ISV, vISV venous ISV. Source data are provided as a Source Data file.

## Circumferential actin formation is associated with ISV constriction

To better understand cortical actin organisation in vessel constriction, we imaged ISVs at higher resolution using the EGFP-UCHD line and identified three major cortical actin organisations: circumferential (C), mesh (M), and longitudinal (L) along with mixed states (C-M and M-L) (Fig. 4a). At 2 dpf, both aISVs and vISVs displayed a heterogeneous mixture of these patterns, with ~50% of the aISVs exhibiting C/C-M actin and ~20% of vISVs displaying C-M actin. From 3 dpf onward, the proportion of vessels with C/C-M actin decreased, whereas the proportion of vessels with M-L/L actin became increasingly common and by 4 dpf predominated in both aISVs and vISVs (>90% of vessels) (Fig. 4b). Thus, circumferential actin is most prominent at early stages, coinciding with the period of marked EC deformation (Fig. 2c, e).

Closer inspection of circumferential actin dynamics revealed that the bundles could appear in multiple configurations: anchored to the plasma membrane (Fig. 4c and Supplementary Movie 5); connected to two cell-cell junctions (Fig. 4d and Supplementary Movie 6); or linking a junction and a membrane on the opposing side of the cell (Fig. 4e and Supplementary Movie 7). In each case, the formation of circumferential bundles coincided with inward deformation of the cell boundary. To test whether circumferential actin correlates with vessel constriction on short time scales, we performed fast time-lapse imaging at 1 min intervals. Regions enriched in circumferential actin (Fig. 4f, grey rectangle) consistently exhibited inward cell deformation and local decreases in vessel diameter (Fig. 4e, f). When circumferential actin dissipated and transitioned to mesh actin, vessel diameter gradually increased, indicating that mesh actin correlates with vessel widening (Fig. 4f and Supplementary Fig. 4). Notably, intermediate states in which circumferential and mesh actin interchanged confirmed that both C and C-M patterns (whenever C bundles were present) correlate with vessel deformation, and that their dynamic alternation produces diameter oscillations consistent with those observed in earlier time-lapse analyses (Fig. 3b). Longitudinal actin was associated with only mild decreases in diameter (Supplementary Fig. 4).

Together, these results demonstrate that cortical actin remodels dynamically during ISV remodelling and that circumferential actin correlates most strongly with vessel constriction, while mesh and longitudinal patterns do not coincide with marked narrowing.

## Circumferential actomyosin bundles generate tensile forces that drive vessel constriction

To test whether circumferential actin mediates contractility, potentially generated by non-muscle myosin II (hereafter, referred to as myosin II), to drive vessel constriction, we used the transgenic lines EGFP-UCHD and *Tg(6xUAS:myl9b-mCherry)$^{rk32}$* (hereafter referred as myl9b-mCherry) to visualise actin and myosin II, respectively. Static imaging revealed myosin II puncta colocalised with actin across circumferential, mesh, and longitudinal organisations. Linear alignment of myosin II along actin bundles was observed in both circumferential and longitudinal organisations (Supplementary Fig. 5a, red arrowheads), but was more frequent and continuous in circumferential bundles. By contrast, mesh actin was mainly associated with punctate myosin II, which was most often positioned between the gaps of the mesh network (Supplementary Fig. 5a, yellow arrowheads).

Time-lapse imaging confirmed these spatial associations. Linear colocalisation of myosin II with circumferential bundles coincided with local vessel constriction, and 3D rendering further verified their spatial overlap (Fig. 5a–c, grey region in kymograph; Supplementary Movie 8, 9). By contrast, lower intensity punctate distributions of myosin II were accompanied by little diameter change or local widening (Fig. 5c, white region in kymograph). Analyses of additional regions demonstrated that circumferential actomyosin consistently correlated with stronger

diameter reduction than other actomyosin patterns (Supplementary Fig. 5b).

To further evaluate the mechanism underlying vessel constriction, simulations were performed using a two-dimensional coarse-grained molecular dynamics model of an actomyosin system, including actin filaments, myosin motors, and passive crosslinkers, enclosed within a deformable cell membrane. In the model, dynamic processes such as actin polymerisation, depolymerisation, and active force generation by motors are present. To promote bundle formation in the circumferential direction, few (less than 1% of the total filaments) actin filaments (2 μm in length, approximately one-fourth of the vessel width) were anchored along the longitudinal membrane. Under suitable parameter conditions, circumferential actomyosin bundles were observed to form, bridging the anchored filaments on opposite sides (Supplementary Fig. 5c and Supplementary Movie 10). The circumferential stress ($\sigma_{xx}$) was found to exceed the longitudinal component ($\sigma_{yy}$) indicating that circumferential bundles promoted the contractile activity in the circumferential direction (Supplementary Fig. 5d). In contrast, when the anchored filaments were introduced along the midline, longitudinal bundles were formed (Supplementary Fig. 5e and Supplementary Movie 11). In this situation, the longitudinal stress ($\sigma_{yy}$) exceeded the circumferential stress ($\sigma_{xx}$) (Supplementary Fig. 5f). Thus, anchoring filaments at different positions can control bundle orientation and consequently the direction of stress.

Upon the introduction of membrane deformability, the circumferential contractile stress indeed reduced the vessel width near the region where bundles were formed. The bundle formation was seen from 16 to 23 mins (Fig. 5d, grey rectangle, and Supplementary Movie 12), which can be identified by the increase in circumferential alignment $N_x^2$ at the central region (CR) (Fig. 5e top, see method for the definition of circumferential alignment). The bundle formation led to the generation of contractile stress in the circumferential direction (Fig. 5e middle), which in turn resulted in a clear reduction in the vessel width (Fig. 5e bottom). Similar behaviour was repeatedly observed over time (Fig. 5e), which was confirmed by the negative correlation between vessel width and circumferential alignment $N_x^2$ at CR with a time lag of about 2.5 mins (Supplementary Fig. 5g). These results indicate a causal relationship between circumferential bundle formation and vessel constriction.

Guided by this simulation, we directly tested whether the different actin organisations bear tensile forces by performing laser ablation using a multiphoton microscope. Ablation on circumferential bundles resulted in the strongest recoil, rapidly producing gaps in the cortical network, whereas longitudinal and mesh bundles showed weaker and comparable recoil responses (Fig. 5f and Supplementary Movie 13–15).

Collectively, these findings demonstrate that circumferential actin, reinforced by myosin II alignment, bears the strongest tensile forces, thereby driving EC contraction and vessel constriction during the early phase of vessel remodelling (2-3 dpf). At later stages, when circumferential bundles are less frequent, mesh and longitudinal actin contribute to cortical tension that helps maintain the reduced vessel diameter.

## Skewing actin organisation in endothelial cells modulates vessel constriction

Since circumferential actin correlated strongly with vessel constriction, we next asked whether altering the balance of actin organisations would impact vessel diameter. We overexpressed Fascin1a, an actin-bundling protein that promotes parallel filament assembly. A plasmid encoding *6xUAS:fascin1a-T2A-mKate2CAAX* was selectively expressed in ECs of EGFP-UCHD embryos (Fig. 6a). Compared to adjacent mKate2CAAX-negative internal controls, Fascin1a overexpression (OE) significantly increased the frequency of aISVs exhibiting circumferential or C-M actin at 2 dpf (C/C-M vs other AOs, Fisher's exact test, $p = 0.04$), with a trend toward more C-M at 3 and 4 dpf (Fig. 6a, b). A

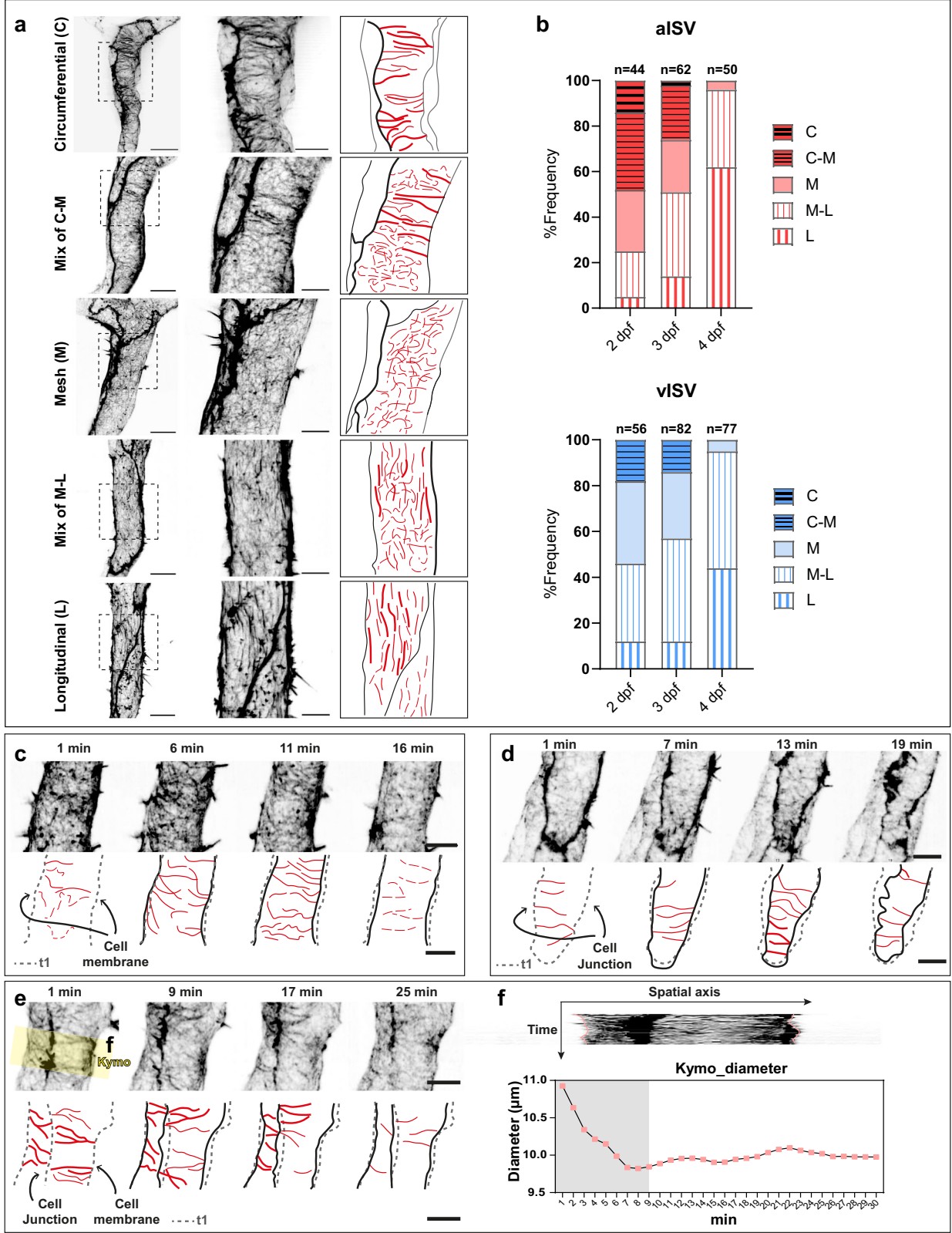

**Fig. 4 | Endothelial cells exhibit cortical actin remodelling, with circumferential actin formation correlating with vessel constriction. a** Representative images of circumferential, mesh, longitudinal and mixed actin organisations at 2 dpf. Scale bars: 10 µm (left), 5 µm (middle). **b** Percentage of each actin organisation from 2 to 4 dpf in aISVs and vISVs. Total number of ISVs (from 22/25/28 embryos at 2/3/4 dpf, 5 experiments) is indicated on the top of the bar. **c**–**e** Still frames from time-lapse movies (Supplementary Movies 5–7) at 2 dpf illustrating circumferential actin anchored at cell membrane (**c** aISV), connected to cell-cell junctions (**d** vISV), or linked to both the membrane and cell junctions (**e** aISV). In the schematics, circumferential actin is highlighted in red. Black lines trace the cell boundary at the current time point, and grey serrated lines represent the outline of the cell membrane or cell-cell junctions at the first time frame. Scale bar, 5 µm. **f** Kymograph of the yellow ROI in (**e**) and measurements of diameter over time. ISV intersegmental vessel, aISV arterial ISV, vISV venous ISV. Source data are provided as a Source Data file.

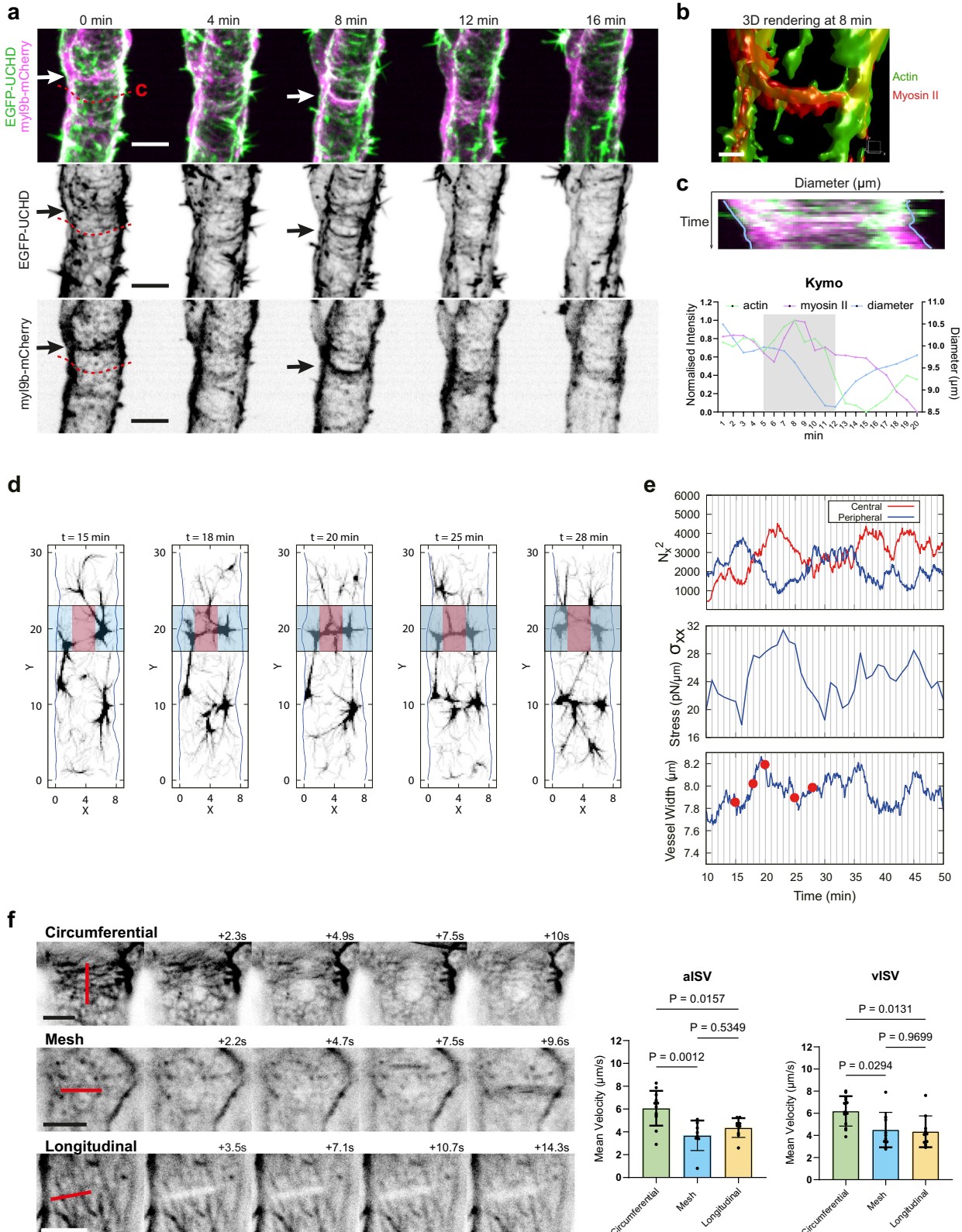

similar trend was observed in vISVs, although the increase at 2 dpf was not significant ($p = 0.27$). This enrichment of C/C-M patterns translated into greater constrictive capacity: between 2-3 dpf, Fascin1a-OE vessels constricted more strongly than controls (~21% aISV, ~22% vISV vs ~ 11% and ~ 9% in controls, respectively), and by 3-4 dpf both vessel types were significantly smaller than controls (Fig. 6c, d) indicating that increasing the proportion of vessels with circumferential actin

enhances constriction. Notably, circumferential actin enrichment was highest at 2 dpf, whereas statistically significant diameter constriction was detected only at 3 and 4 dpf.

Conversely, to test the effect of enhancing mesh actin, we over-expressed WASb (human homologue, Wasp), a key regulator of actin nucleation that facilitates the formation of branched actin networks[9,21]. Similarly, we injected a plasmid encoding *6xUAS:wasb-mCherry* into

**Fig. 5 | Circumferential actomyosin bundles generate highest tensile forces during vessel constriction. a** Still images (Supplementary Movie 8) of an aISV at 2 dpf showing actin (EGFP-UCHD) and non-muscle myosin II (myl9b-mCherry) from *Tg(fli1:GAL4FF)^ubs3; Tg(UAS:EGFP-UCHD)^ubs18* and *Tg(6xUAS:myl9b-mCherry)^rk32*. Linear colocalisation of circumferential actin and myosin II is observed at 8 min. Scale bar: 5 μm. **b** 3D rendering (Supplementary Movie 9) of actomyosin colocalisation at 8 min showed in (**a**). Scale bar, 1 μm. **c** Kymograph of region indicated in (**a**) and quantification of normalised $\log_{10}$ fluorescence intensity of UCHD and myl9b, and corresponding vessel diameter. The grey rectangle indicates the time range when linear colocalisation of actomyosin appears. **d** Snapshot images of filament density obtained by simulation. Blue rectangles indicate circumferential actin formation; red rectangles indicate the central region of the bundle. **e** Temporal evolution of circumferential actin alignment ($N_x^2$) at the central (red) and peripheral (blue) regions (top), circumferential stress (middle), and vessel width (bottom). Snapshot time points in (**d**) are marked as red dots in (**e**) (bottom). **f** Representative images showing laser ablation (red line) on circumferential, mesh and longitudinal actin organisations at 2 dpf (Supplementary Movie 12–14). Scale bars, 5 μm. Mean velocity from the first 10 s was measured (aISV $n = 12/8/9$ and vISV $n = 13/9/10$ in C/M/L, from 27 embryos in 3 experiments). Statistical significance was assessed by ordinary one-way ANOVA with Tukey's multiple comparisons test. Data are shown as mean ± SD. ISV intersegmental vessel, aISV arterial ISV, vISV venous ISV. Source data are provided as a Source Data file.

EGFP-UCHD embryos (Supplementary Fig. 6a, c). Wasb-OE increased the frequency of ISVs exhibiting mesh actin, interestingly, at the expense of longitudinal rather than circumferential actin (Supplementary Fig. 6a, b). In both aISVs and vISVs, this shift was most apparent at later developmental stages (at 4 dpf, M vs other AOs, aISV, Fisher's test aISV, $p = 0.04$; vISV, $p = 0.02$). Correspondingly, Wasb-OE vessels were dilated compared to controls (Supplementary Fig. 6c, d), yet continued to constrict over time to a degree comparable to controls, indicating that increasing mesh actin preserved relative narrowing despite larger absolute diameters.

### Myosin II inhibition blocks endothelial cell contraction and reduces vessel constriction

To investigate the contribution of myosin II activity on EC contraction and vessel constriction, we overexpressed a plasmid encoding dominant-negative myosin light chain 9b (*6xUAS:myl9bA2A3-EGFP*) in *Tg(fli1:GAL4FF)^ubs3; Tg(fli1:myr-mCherry)^ncv1* and compared mosaic overexpression cells with adjacent GFP-negative internal controls (Fig. 6e). aISVs with Myl9bA2A3-OE ECs, and therefore reduced myosin II activity, were consistently wider compared to controls from 2 to 4 dpf, whereas vISVs showed increased diameters at 2 and 4 dpf but not at 3 dpf (Fig. 6f).

In Myl9bA2A3-OE cells, cell area remained unchanged from 2 to 4 dpf (Supplementary Fig. 6f), unlike the reduction observed in controls (Fig. 2b), indicating impaired contraction. Normally, aECs narrowed, and aISVs constricted (Fig. 2c), but with myosin II inhibition, aECs enlarged along both radial and axial axes at 2-3 dpf. Radially, vessel constriction was partly preserved by enhanced radial-to-axial rearrangement (negative cell number strain), even as cells widened. Axially, elongation arose primarily from cell lengthening (positive axial cell strain) with little cell addition. Consistent with this, in Myl9bA2A3-OE vessels, total EC number increased between 2 to 4 dpf, but lacked the stepwise rise seen in controls, which showed significant growth between 2-3 and 3-4 dpf (Supplementary Fig. 6f), suggesting delayed or uncoordinated cell addition during vessel remodelling. By 3-4 dpf, continued cell expansion in the radial axis overwhelmed axial cell rearrangement, resulting in slight vessel widening, while elongation was recovered through cell addition (Fig. 6h).

In control vISVs, high proliferation at 2-3 dpf increased circumferential coverage, but cell narrowing offset vessel widening; at 3-4 dpf, axial rearrangement (negative radial cell number strain) drove constriction (Fig. 2e). Under myosin II inhibition, vISVs exhibited less radial EC narrowing compared to control, and along with enhanced radial-to-axial cell rearrangement (negative radial cell number strain) at 2-3 dpf, leading to modest constriction (negative radial vessel strain) (Fig. 6i), in contrast to the slight widening observed in controls (Fig. 2e). By 3-4 dpf, ECs reversed to radial expansion, offset the effects of axial rearrangement, leading to modest constriction, which resulted in wider vessels than controls (Fig. 6f). Axial elongation during 2-3 dpf was driven primarily by cell addition (positive axial cell number strain) despite cell shortening (Supplementary Fig. 6g), whereas at 3-4 dpf elongation was modest and supported by cell lengthening (Fig. 6i).

Despite these changes, overall vessel length remained comparable to controls (Supplementary Fig. 6h).

Together, these findings demonstrate that myosin II activity is important for coupling EC contraction with rearrangement, and that its inhibition leads to enlarged ECs, disrupted rearrangement and exchange, and ultimately impaired vessel constriction.

### Krit1 promotes circumferential actin assembly, cell contraction and vessel constriction

Since our results showed that circumferential actomyosin contractility are essential for EC contraction and vessel constriction, we next asked whether defects in these processes underlie CCM. Loss-of-function mutation in *ccm1/krit1* is known to cause enlarged, poorly remodelled vessels in zebrafish and mammals[16,17], but the underlying cellular mechanisms remain unclear. Krit1 has been reported to localise at endothelial junctions, where it interacts with junctional complexes and cytoskeletal elements, suggesting it may scaffold cortical actin[22]. We confirmed in zebrafish ISVs that Krit1 is enriched at junction-cortex interfaces and colocalised with circumferential actin, bridging cell-cell junctions and cortical bundles (Fig. 7a). This observation suggests that Krit1 may anchor actin bundles at cell-cell junctions to organise cortical actin. To test this, we performed mosaic endothelial overexpression of Krit1 (mStayGold-Krit1) and compared vessels comprising Krit1-OE cells with adjacent mStayGold (mSG) -negative internal controls (Fig. 7b). Krit1-OE increased the proportion of ISVs displaying circumferential and C-M actin at 2 and 3 dpf (Fig. 7c, C/C-M vs other AOs at 2 dpf, aISV, Fisher's exact test $p = 0.04$; vISV, $p = 0.02$) and enhanced vessel constriction during this period (aISV ~-12% vs -0.5%; vISV ~-9% vs ~-7%), resulting in reduced aISV diameter at 3 and 4 dpf and vISV diameter at 2 and 3 dpf (Fig. 7d).

We next examined a zebrafish model of CCM containing a *krit1* loss-of-function mutation (*krit1^t26458*, referred to as *krit1^-/-* hereafter). By transplanting cells from embryos obtained from in-crosses of *krit1^+/-; Tg(fli1:GAL4FF)^ubs3; Tg(UAS:EGFP-UCHD)^ubs18* zebrafish (yielding *krit1^+/+*, *krit1^+/-*, and *krit1^-/-* genotypes) into wild-type recipients, we generated mosaic ISVs that experienced normal blood flow. This allowed us to investigate cell autonomous role of Krit1 on actin organisation, cell deformation and vessel constriction. Clear differences were observed between genotypes. In wild-type and heterozygous grafts, circumferential and C-M actin were present from 2 to 3 dpf, supporting balanced cell shape control. In *krit1^-/-* grafts, circumferential actin was strongly reduced with a shift towards longitudinal arrangements in both aISVs and vISVs (Fig. 7e, f, ML/L vs other AOs at 2dpf, aISV, Fisher's exact test $p = 0.0007$; vISV, $p = 0.02$). Consistently, vessel diameters were larger in *krit1^-/-* aISVs at 2, 3 and 4 dpf and vISV at 3 and 4 dpf compared to wild types, suggesting defects in vessel constriction (Supplementary Fig. 7a, b). Vessel lengths were comparable across groups except for vISVs at 2dpf, indicating that the primary defect lays in radial rather than axial growth (Supplementary Fig. 7c).

Analysis of cell size and strain revealed the cellular basis of this defect. Cell area in homozygous mutants remained similar to controls at 2 dpf but increased significantly at 3 dpf in vECs and in both

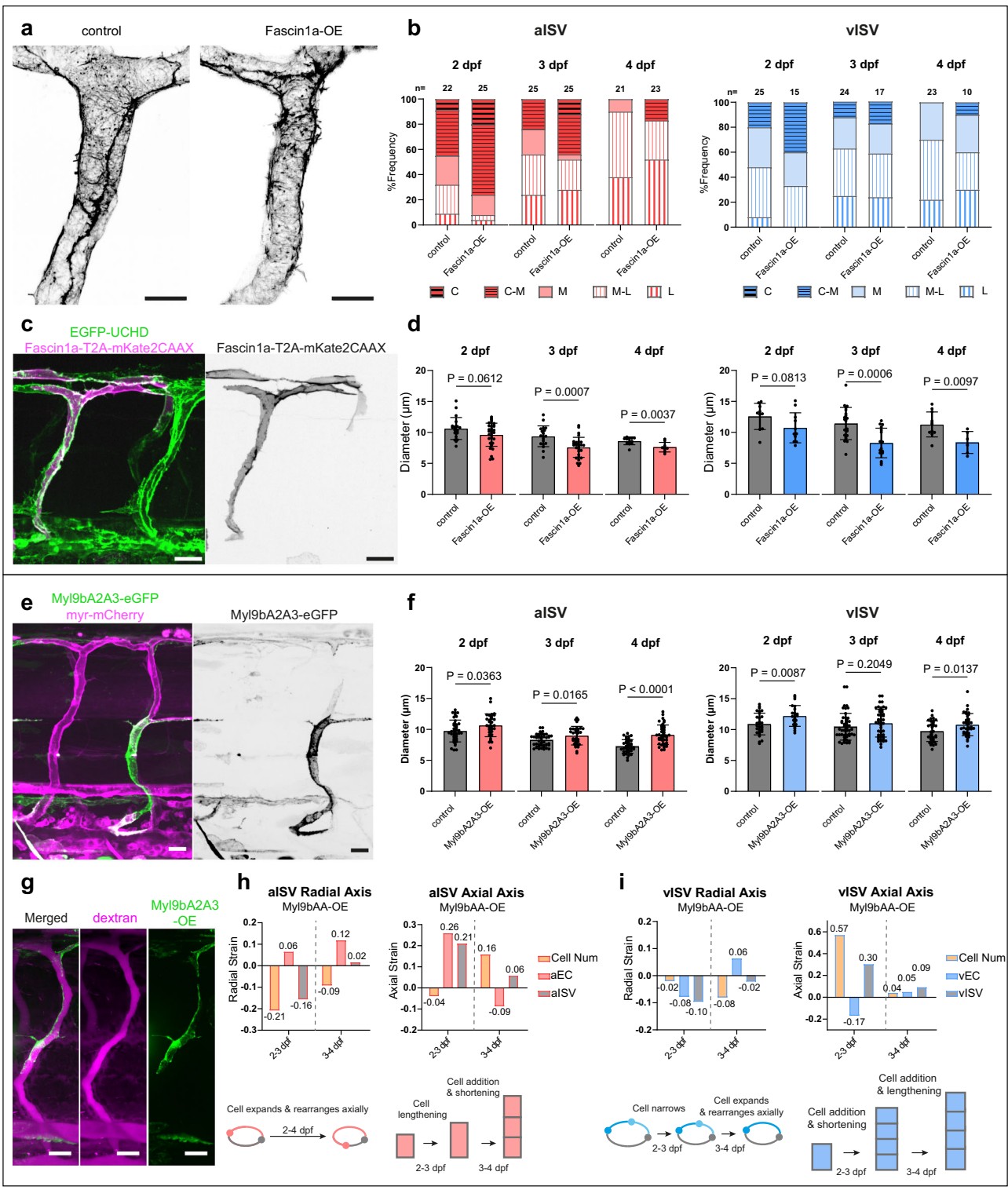

cell types by 4 dpf, whereas control cells progressively contracted (Supplementary Fig. 7d). In control aISVs, EC contraction along radial axis promoted constriction despite cell influx between 2 and 3 dpf, and combined with axial rearrangement of cells between 3 and 4 dpf, maintained reduced diameter (Fig. 7g). By contrast, *krit1*[-/-] cells expanded rather than contracted between 2 and 4 dpf, leading to vessel dilation even as some cells left the circumference. *krit1*[-/-] cells also lengthened axially between 3 and 4 dpf, rather than shortening, reinforcing overall enlargement (Supplementary Fig. 7e). In control vISVs, vessel constriction between 2 and 3 dpf

was achieved by counterbalancing the addition of cells with cell contraction, and between 3 and 4 dpf was maintained primarily through axial rearrangement (wild type) or rearrangement plus contraction (heterozygotes) (Fig. 7h). In *krit1*[-/-] vISVs, however, cells failed to contract and instead expanded radially, so vessel strain shifted toward reduced or even positive values. At the same time, additional cells were accommodated axially, reflected by elevated axial strain (Supplementary Fig. 7f). Thus, without Krit1, ECs lose the coordinated contraction and rearrangement that normally drive vISV constriction.

**Fig. 6 | Vessel constriction is enhanced by Fascin1a-induced circumferential actin assembly but disrupted by decreased Myosin II activity.**
**a, c** Representative images of ISVs of 2 dpf (*Tg(fli1:Gal4ff)^{ubs3};Tg(UAS:EGFP-UCHD)^{ubs18}*) embryos with or without Fascin1a-T2A-mKate2CAAX overexpression (internal negative control). Scale bars, 10 μm (**a**) and 20 μm (**c**). **b** Actin organisation in control and Fascin1a-OE vessels at 2-4 dpf. Total number of ISVs (from 25/26/23 embryos at 2/3/4 dpf, 7/6/6 experiments) is indicated on the top of the bar. Fisher's exact test (C/C-M vs other AOs, control vs Fascin1a-OE: aISV, *p* = 0.04/0.23/0.10; vISV, *p* = 0.27/0.67/0.30 at 2/3/4 dpf. **d** Quantification of vessel diameter in control and Fascin1a-OE aISVs (*n* = 20/19/12 vs 32/29/9) and vISVs (*n* = 10/21/11 vs 11/16/6) at 2/3/4 dpf (10/12/5 embryos; 2/3/2 experiments). Each point represents one ISV. Data are mean ± SD; unpaired two-tailed *t* test. **e** Representative images of ISVs of 2 dpf *Tg(fli1:myr-mCherry)^{ncv1}* embryos expressing constructs encoding dominant

negative myosin light chain protein *6xUAS:myl9bA2A3-eGFP*. Scale bar, 20 μm. **f** Quantification of vessel diameter in control and Myl9bA2A3-OE aISVs (*n* = 39/43/37 vs 32/33/41) and vISVs (*n* = 29/49/33 vs 23/48/36) from 2/3/4 dpf (17/17/20 embryos; 2/2/2 experiments). Each point represents one ISV. Data are mean ± SD; unpaired two-tailed *t* test. **g** Representative images of ISVs of 2 dpf wild-type embryos expressing Myl9bA2A3-eGFP in single cells. Microangiography is performed by injecting dextran rhodamine. Scale bar, 10 μm. **h** Radial and axial strains at 2-3 dpf and 3-4 dpf in myl9bA2A3-OE aISVs (averages from aISV, *n* = 32/33/41, aEC *n* = 24/10/18 at 2/3/4 dpf). **i** Radial and axial strains at 2-3 dpf and 3-4 dpf in myl9bA2A3-OE vISVs (averages from vISV, *n* = 23/48/36, vEC *n* = 12/22/27 at 2/3/4 dpf). Embryos used: *n* = 17/17/20 for vessel strain; *n* = 10/6/12 for cell strain, from 2/2/2 experiments. ISV intersegmental vessel, aISV arterial ISV, vISV venous ISV. Source data are provided as a Source Data file.

To investigate whether these defects also involve impaired acto-myosin coupling, we analysed embryos derived from in-crosses of *krit1^{+/-}; Tg(fli1:GAL4FF)^{ubs3}; Tg(UAS:EGFP-UCHD)^{ubs18}; Tg(6xUAS:myl9b-mCherry)^{rk32}* which generate homozygous mutants lacking blood flow, and quantified junctional and cortical intensities of actin and myosin II. Compared with transplanted *krit1^{-/-}* cells that experience normal circulation, flow-deficient mutants exhibited even larger cell size and prominent cortical actin bundles that followed a diagonal-to-longitudinal direction (Supplementary Fig. 7g). In wild-type embryos, junctional actin was enriched over cortical actin (ratio > 0), whereas this enrichment was significantly reduced in *krit1^{-/-}* cells, consistent with impaired junctional actin assembly (Supplementary Fig. 7h). For myosin II, mean junction/cortex ratios were similar across genotypes (ANOVA, *P* = 0.23), but variance was reduced in mutants (Brown-Forsythe test, *P* = 0.037). Wild-type cells showed heterogeneous distribution, while mutants displayed more uniform patterns, indicating a loss of dynamic heterogeneity that may constrain contractile responses. Thus, Krit1 is required both for junctional actin enrichment and for myosin II distribution.

In sum, we demonstrate that Krit1 promotes the assembly and stabilisation of circumferential actin at junctional sites, enabling EC contraction and vessel constriction. Uncoupling of Krit1 from the actin cytoskeleton results in the impairment of vessel remodelling and culminates in enlarged vessels that are characteristic of CCMs.

## Discussion

During vascular remodelling, zebrafish ISVs undergo a seemingly paradoxical process of radial constriction as EC number increases. We resolved this paradox by showing that circumferential actin bundles, anchored by Krit1 and reinforced by myosin II, provide the contractile machinery that enables ECs to contract and rearrange, narrowing the vessel while accommodating additional cells. When this contractile machinery is compromised, through disruption of actomyosin activity or loss of Krit1, ECs fail to contract, rearrangement is impaired, and vessels dilate instead of constricting. Conversely, enhancing circumferential actin bundle formation augmented vessel constriction. These findings establish circumferential actomyosin assembly as a key mechanism by which EC mechanics directly contribute to vessel size control during development (Fig. 8).

High-resolution timelapse imaging revealed the presence of different actin organisations in the EC cortex, which dynamically remodels. This in vivo temporal analysis of endothelial actin organisation was performed at near in vitro detail. How actin is organised and remodelled during vessel remodelling is not yet clear, but likely depends on specific cytoskeletal regulators. For example, the formin actin nucleator protein, DAAM, helps align actin bundles along the circumferential direction in the Drosophila trachea[23,24]. In zebrafish ISVs, circumferential actin bundles are enriched at 2 dpf, coinciding with peak EC contraction, and functional assays confirmed their

importance: Fascin1a overexpression increased circumferential bundles and enhanced vessel constriction, whereas Wasb overexpression favoured mesh actin and reduced constriction. Importantly, circumferential actin was also observed in larger vessels (DA and the caudal vein plexus), coupled with myosin II (Supplementary Fig. 8a, b and Supplementary Movie 16–18), indicating that this contractile architecture has a broader role in vascular morphogenesis. We also observed converging actomyosin nodes (Supplementary Fig. 8c), resembling actin asters in cultured cells[25]. These findings suggest that ECs deploy multiple actin structures, including circumferential bundles and asters, to achieve constriction depending on vessel context.

While CCM studies have uncovered key signalling pathways that are dysregulated in the disease[4], it is still unclear how altered signalling causes the pathological vessel morphology. We show that Krit1-deficient ECs exhibit altered cortical actin remodelling that hinders cell contraction. Mechanistically, Krit1/CCM1 and CCM2 normally stabilise the connections between junctional proteins (VE-cadherin) and the actin cytoskeleton through interactions with Rap1 and β-catenin, and integrins via ICAP-1[26–28]. This coupling generates tensile forces that contract ECs to narrow vessels. Loss of Krit1 uncouples this scaffold, leading to defective actin organisation and impaired actomyosin contractility. How actomyosin dynamics couple to Krit1 remains debated: some studies report ROCK-driven hypercontractility and increased stress-fibre formation upon Krit1 loss[28–30], while others showed reduced junctional myosin II activity[31]. This discrepancy likely reflects differences in cellular context: earlier work emphasised cortical stress fibres and cell-matrix adhesions in vitro, whereas the latter highlighted junctional contractility in vivo. We observed reduced junctional actin and mis-localised myosin II in mutants, suggesting that Krit1 influences the balance between junctional and cortical actomyosin organisation (Supplementary Fig. 7h). Krit1 is known to differentially control ROCK isoforms, recruiting ROCK2 to junctions while restraining ROCK1-driven ventral actin stress fibres[29]. In wild-type ISVs, circumferential actin bundles connect to junctions, enabling myosin II to transmit contractile forces that drive constriction. In *krit1* mutants, actin shifts toward longitudinal bundles, which are largely disconnected from junctions, and myosin II is diffusely distributed, uncoupled from junctional networks, preventing effective force transmission and EC constriction. Whether longitudinal actin in Krit1-deficient ECs represents stress fibres remains unclear, since we cannot resolve apical versus basal membranes in the zebrafish model. Notably, cell enlargement was more severe in flow-deficient *krit1^{-/-}* cells (from in-cross, Supplementary Fig. 7h) than in cells exposed to flow (from cell transplants, Fig. 7e), suggesting that blood flow may partially mitigate the severity of Krit1 loss-of-function, a cellular mechanism that merits further investigation[17]. Taken together, these data indicate that Krit1 stabilises junctional-cortical circumferential actomyosin scaffolds that drive constriction, and its loss leads to context-specific dilation. Notably, mosaicism in transplanted vessels may modulate the

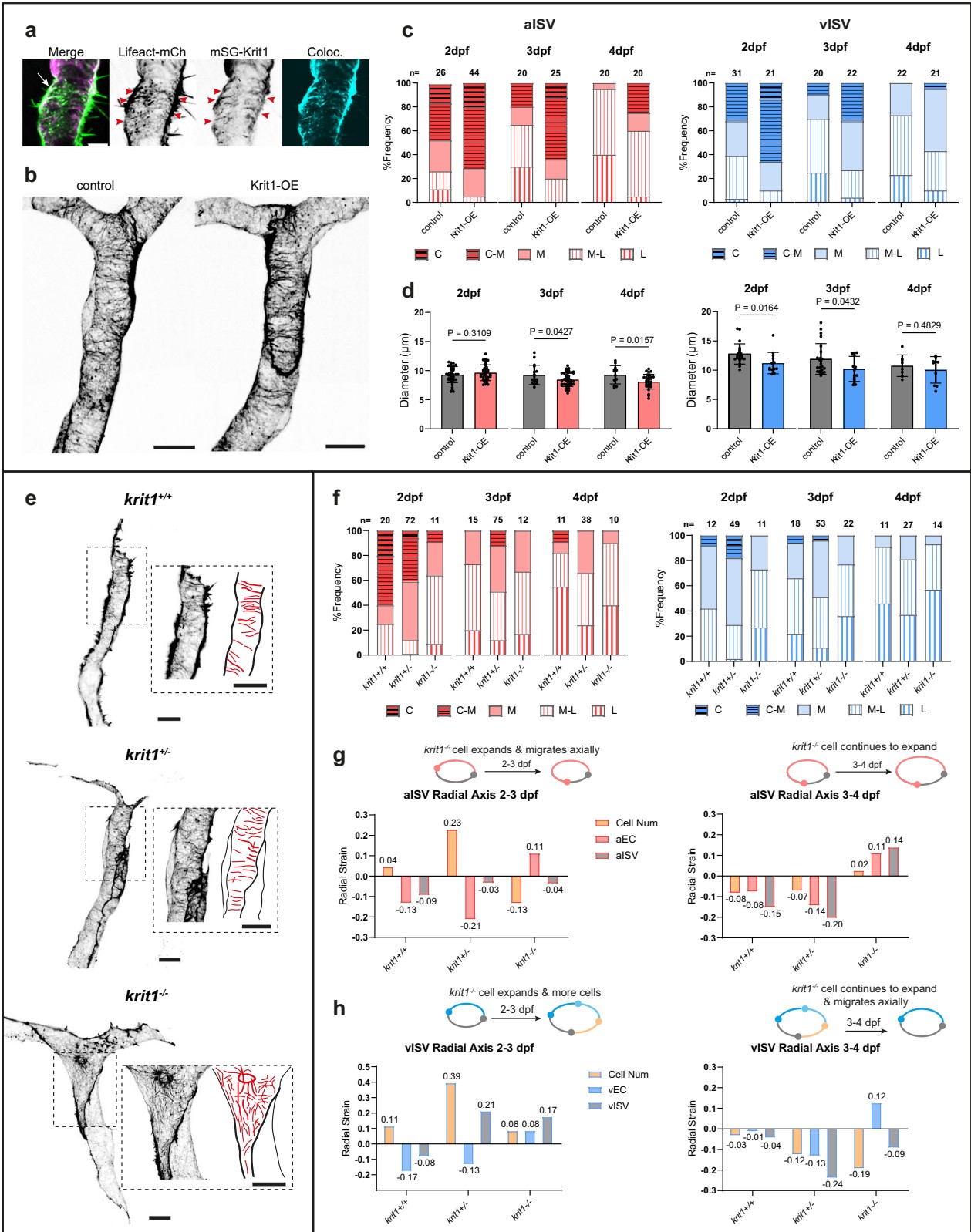

actomyosin organisation of mutant cells through their integration within a wild-type mechanical environment, contributing to differences between mosaic, flow-exposed cells and fully mutant, no-flow vessels.

Apart from cell deformation, endothelial rearrangements also influence vessel size by redistributing cells axially and reducing circumferential occupancy to support radial narrowing. Circumferential

actin may facilitate this process, as bundles were observed connecting junctions at the rear of migrating cells (Fig. 4d). We identified a transient "self-seam" junction, where a single EC forms self-contacts for hours. This intermediate between self-splitting and self-fusion[32] likely stabilises lumen integrity during the highly dynamic phase of remodelling. Mural cells are also implicated in vessel size control, regulating diameter in mature vascular beds[33–36], but analysis of *pdgfrb*[SAI6389]

**Fig. 7 | Failure to assemble circumferential actin bundles in krit1⁻/⁻ cells compromises cell contraction and vessel constriction. a** Colocalisation of circumferential actin with mStayGold-Krit1 in endothelial cortex and junction in a 2 dpf *Tg(fli1:Lifeact-mCherry)* embryo. White arrow, junction; red arrowheads, colocalisation. Scale bar, 5 μm. **b** Representative aISVs at 2 dpf with or without mStayGold-Krit1 overexpression. Actin is visualised using *Tg(fli1:Gal4ff)ubs3; Tg(UA-S:EGFP-UCHD)ubs18* zebrafish. Scale bars, 10 μm. **c** Frequency of control and Krit1-OE vessels with different actin organisations at 2-4 dpf (ISV numbers shown above bars; 44/32/27 embryos at 2/3/4 dpf; 3/3/3 experiments). Fisher's exact test (C/C-M vs other AOs, control vs Krit1-OE): aISV, *p* = 0.04/0.006/0.04; vISV, *p* = 0.02/0.13/0.48 at 2/3/4 dpf. **d** Quantification of vessel diameter in control and Krit1-OE aISVs (*n* = 37/17/12 vs 33/41/29) and vISVs (*n* = 22/23/8 vs 13/15/10) at 2/3/4 dpf (27/21/18 embryos; 3/3/3 experiments). Mean ± SD, unpaired two-tailed *t* test. **e** Representative images of *krit1+/+*, *krit1+/-* and *krit1⁻/⁻* transplanted ECs in wild-type embryos at 2 dpf showing different actin organisations (highlighted in red in

schematics). Actin is visualised using EGFP-UCHD zebrafish. Scale bar, 10 μm. **f** Percentage of aISVs and vISVs showing actin organisation between *krit1+/+*, *krit1+/-* and *krit1⁻/⁻* from 2 to 4 dpf. ISV numbers shown above bars (*krit1+/+*: 11/15/11; *krit1+/-*:38/36/28; *krit1⁻/⁻*:12/12/12 embryos at 2/3/4 dpf, 15/18/17 experiments). Fisher's exact test (ML/L vs other AOs, among *krit1+/+*, *krit1+/-* and *krit1⁻/⁻*): aISV, *p* = 0.0007/0.22/0.30; vISV, *p* = 0.02/0.08/0.65 at 2/3/4 dpf. **g** Radial strains at 2-3 dpf and 3-4 dpf in aISVs consisting of *krit1+/+* (aISV, *n* = 8/9/12; aEC *n* = 12/14/17 at 2/3/4 dpf), *krit1+/-* (aISV, *n* = 21/11/8, aEC *n* = 23/18/14 at 2/3/4 dpf), and *krit1⁻/⁻*(aISV, *n* = 11/8/6, aEC *n* = 20/16/11 at 2/3/4 dpf) cells. **h** Radial strains at 2-3 dpf and 3-4 dpf in vISVs consisting of *krit1+/+* (vISV, *n* = 6/6/10; vEC *n* = 7/12/14 at 2/3/4 dpf), *krit1+/-* (vISV, *n* = 14/7/9, vEC *n* = 25/17/20), and *krit1⁻/⁻* (vISV, *n* = 14/13/13, vEC *n* = 17/19/23) cells. Embryos used for both ISVs: *krit1+/+* (*n* = 6/8/11), *krit1+/-* (*n* = 9/8/9), and *krit1⁻/⁻* (*n* = 5/5/6) for vessel strain, from 6/7/9 experiments at 2/3/4 dpf; *krit1+/+* (*n* = 6/8/12), *krit1+/-* (*n* = 9/8/8), and *krit1⁻/⁻* (n = 7/5/7) for cell strain, from 12/11/12 experiments at 2/3/4 dpf. Source data are provided as a Source Data file.

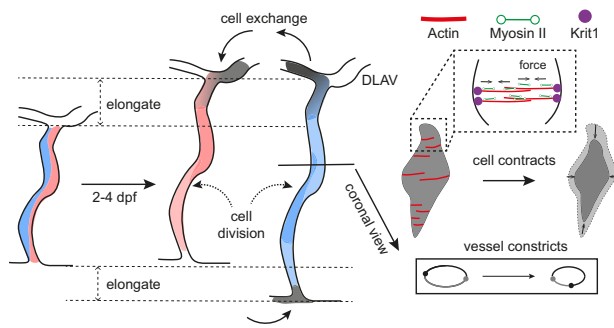

**Fig. 8 | Graphical model of vessel size control during vascular remodelling.** Between 2 and 4 dpf, zebrafish ISVs elongate while simultaneously undergoing radial constriction, even as EC numbers increase through division and cell exchange. This paradox is resolved by circumferential actin bundles, anchored by Krit1/CCM1 at cell-cell junctions and reinforced by myosin II, which generate contractile forces that drive EC contraction and rearrangement. These coordinated cytoskeletal mechanisms allow vessels to narrow while accommodating additional cells. When actomyosin activity or Krit1 function is disrupted, ECs fail to contract and rearrange, resulting in vessel dilation. This model highlights how EC mechanics, through the integration of actin, myosin II and Krit1, regulate vessel diameter during vascular remodelling.

mutants with reduced pericyte coverage[37] showed normal ISV constriction between 2 and 4 dpf (Supplementary Fig. 9a–c), indicating that early vessel narrowing relies on endothelial-intrinsic mechanisms.

Venous ISVs, where ECs proliferate more frequently, may generate compressive forces that compact cells[30] and augment shrinkage. Indeed, vECs shrank more than aECs despite showing less circumferential actin (Fig. 2e, f). Inhibition of proliferation with aphidicolin (Aph) and hydroxyurea (HU)[38] reduced EC numbers, but the effect was similar in both aISVs and vISVs (Supplementary Fig. 10a, b), preventing isolation of venous-specific contributions. Methods that more specifically target venous cell proliferation are required in future studies. Nonetheless, while cell crowding from proliferation may contribute to cell compaction, our laser ablation experiments confirm that active circumferential contraction provides tensile force for vessel constriction.

The transition from circumferential to longitudinal actin may be influenced by hemodynamic forces. In vitro, increased luminal pressure in 3D tubes promotes circumferential alignment[39], whereas shear stress aligns bundles with flow direction in cultured ECs[40]. In zebrafish ISVs, circumferential actin peaks at 2 dpf when cell contraction is strongest, despite rising flow velocity and wall shear stress between 2-3 dpf and declining afterwards[41]. This divergence from in vitro observations suggests that pressure, stretch and shear act in combination

in vivo, with outcomes also influenced by endothelial state (migratory, proliferative, or quiescence).

A limitation of our study is that actin organisation was classified manually. Although our attempted automated classification confirmed the overall circumferential-to-longitudinal trends, it failed to reliably capture mixed classes (Supplementary Fig. 11a–c and Supplementary Note 2). The automated approach only identified isotropic orientations and did not distinguish between mesh and C-L, or mixtures of C-M and M-L, due to a global analysis. Such analysis potentially requires segmentation of thin actin fibres, which were not visible due to low intensity signals and low image resolution. We therefore relied on manual annotation as the most consistent approach to capture the complexity in actin organisation. Another consideration is the mosaic nature of our overexpression and transplantation experiments, which, while powerful for the investigation of individual EC behaviours and function, may introduce variability in cell-cell interactions within chimeric vessels. This is particularly relevant in CCM mutants, where mutant cells influence neighbouring cells[42–44]. Despite these caveats, our multiscale approach, from cytoskeletal organisation to vessel-level mechanics, provides a comprehensive framework for how endothelial actomyosin drives vessel constriction, and how its disruption contributes to pathological dilation in CCM.

## Methods

### Zebrafish husbandry and manipulation

All animal experiments were approved by the Institutional Animal Care and Use Committee at RIKEN Kobe Branch (IACUC). Zebrafish (*Danio rerio*) were maintained and raised following standard protocols. The transgenic lines used in this study included *Tg(fli1:MYR-EGFP)ncv2* [45], *Tg(fli1:h2bc1-mCherry)ncv31* [46], *Tg(fli1a:H2B-EGFP)ncv69* [47], *Tg(fli1:Lifeact-mCherry)ncv7* [45], *Tg(fli1:myr-mCherry)ncv1* [48], *Tg(fli1:GAL4FF)ubs3* [49], *Tg(UA-S:EGFP-UCHD)ubs18* [50], *Tg(fli1ep:Lifeact-EGFP)zf495* [51], *Tg(6xUAS:myl9b-mCherry)rk32* (this study), while the mutant lines used were *krit1t26458* [16] and *pdgfrbSA16389* [37]. Fish embryos were collected within 30 minutes of divider removal and were allowed to develop at 28.5 °C to the appropriate stage. For genotyping of the *krit1t26458* mutant, genomic DNA was extracted using the HotSHOT method, and PCR amplification was performed with specific primers (Forward 5'-CCA-CAAGCGTAACGTAAATG; Reverse 5'-ATCTATGGACGCAATGCAG). The resulting PCR products were purified and sequenced using the forward primer. Genotyping of the *pdgfrbSA16389* mutant was performed as previously described[37].

### Plasmid production and injection

Plasmids used in this study were constructed using either NEBuilder® HiFi DNA Assembly or In-Fusion® HD Cloning kits, following the respective manufacturer protocols. The *wasb-mCherry* vector[9] (a gift

from Holger Gerhardt, Max Delbrück Centre for Molecular Medicine, Germany) and the *Fscn1a-T2A-mKate2CAAX* vector[12] were amplified by PCR and incorporated into Ac/Ds transposon-based constructs. The plasmid encoding Krit1[17] (a gift from Salim Abdelilah-Seyfried, Potsdam University, Germany) was amplified and fused with mStayGold and cloned into Ac/Ds constructs. The *myl9bA2A3-eGFP* vector[52] (a gift from Anne Schmidt, Institute Pasteur, France) was amplified by PCR and assembled into Tol2 transposon-based constructs. Assembly reactions employed the appropriate backbone vectors and primers as specified by the cloning protocols. For details on vectors, refer to Supplementary Data 1. For plasmid overexpression experiments, embryos were injected at the one-cell stage with 2 nl of a mixture containing 50-100 ng of plasmid DNA, 100-200 ng Tol2 or Ac transposase mRNA, phenol red for tracking, and Milli-Q water to a final volume.

## Imaging

Embryos were immobilised in 0.8% low-melt agarose (Bio-Rad) within E3 medium supplemented with 0.16 mg/mL Tricaine and 0.003% phenylthiourea to reduce movement and pigmentation, respectively. Confocal imaging was performed using an inverted Olympus IX83 microscope, equipped with a Yokogawa CSU-W1 spinning disk confocal unit. Imaging was carried out with an Olympus UPLSAPO x 60/NA 1.2 water immersion objective, × 40/NA 1.25 or × 30/NA 1.05 silicone oil immersion objectives and a Zyla 4.2 CMOS camera (Andor). Image acquisition was controlled using Andor iQ3 software, and z-stack images were captured for 3D reconstructions. Actin organisation was imaged using two advanced microscopy setups. The first system was an Andor/Olympus Dragonfly 200 inverted microscope equipped with a 60x water immersion objective lens (Olympus UPLSAPO, NA 1.2) and a Zyla 4.2 PLUS sCMOS camera (2.0x magnification). Imaging utilised a 40 μm pinhole, and z-stacks were acquired at Nyquist intervals. Imaging parameters were controlled via Fusion Control software (2.3.0.50). The second system was a customised spinning disk super-resolution microscope (SDSRM)[53], implemented based on the commercial IXplore IX83 SpinSR system (Evident), equipped with a 60x silicon immersion objective lens (Evident UPLSAPO 60XS, NA 1.3). z-stack interval was 0.15um.

## Microangiography

Microangiography was performed for single-cell analysis. Embryos at 2, 3, and 4 dpf were injected with 1-2 nL of dextran tetramethyl rhodamine (molecular weight 2000 kDa, Invitrogen) at a concentration of 10 mg/mL. The dextran solution was introduced through the duct of Cuvier of the zebrafish. Imaging was performed immediately following the injection. Confocal z-stacks were acquired using an Olympus UPLSAPO 60x/1.2 NA water immersion objective.

## Cell transplantation

Donor cells from embryos derived from *krit1*[+/- t26458] in-crossed zebrafish in *Tg(fli1:GAL4FF)*[ubs3]; *Tg(UAS:EGFP-UCHD)*[ubs18] background were collected from random locations in the blastoderm. These cells were transplanted into the lateral marginal zone of the blastoderm in wild-type recipient embryos at 4.5–6 hpf. The procedure was carried out using a CellTram® 4r Oil microinjector (Eppendorf) with a borosilicate glass capillary GC100-15 (Harvard Apparatus Ltd). The capillary tips were shaped into a smooth spoon using a horizontal micropipette puller P-87 (Sutter Instrument Co.), and an MF-900 microforge (Narishige) for precision. During transplantation, embryos were maintained in agarose-coated Petri dishes containing 0.5 x E2 buffer composed of 7.5 mM NaCl, 0.25 mM KCl, 0.5 mM MgSO₄, 75 μM KH₂PO₄, 0.5 mM CaCl₂, 0.35 mM NaHCO₃, and 50 μ/mL penicillin-streptomycin. Recipient embryos were grown to 2, 3, and 4 dpf, and those exhibiting EGFP-positive ISVs were selected for microangiography and imaging. Genotyping was performed on the

corresponding donor embryos. A total of 24 independent transplantation experiments was conducted.

## Image processing

Raw images of actin organisation captured using the Dragonfly 200 microscopy were deconvoluted with ™Huygens Spinning Disk Deconvolution, following the standard protocol established in the unit. The deconvoluted images were subsequently processed using Fiji software (NIH). To separate the front and back sides of the vessel, the ISV was first straightened using Fiji's *Straighten* tool and then divided with the *Split* tool. Z-stacks for each half were projected into single images using *maximum intensity projection*. To enhance consistency and reproducibility, a custom macro was developed in Fiji to semi-automate the straightening and splitting process. The spline of each ISV was manually traced for straightening, and the central plane of the z-stack was manually selected for splitting (see Supplementary Fig. 2).

## Quantification of blood vessel diameter and length and EC number

To quantify vessel diameter and length, live *Tg(fli1:MYR-EGFP)*[ncv2], *Tg(fli1:myr-mCherry)*[ncv1], or *Tg(fli1:GAL4FF)*[ubs3]; *Tg(UAS:EGFP-UCHD)*[ubs18] transgenic embryos were imaged at 2, 3, and 4 dpf. For EC number, live *Tg(fli1:h2bc1-mCherry)*[ncv31] or *Tg(fli1a:H2B-EGFP)*[ncv69] embryos were imaged at the same stages. Confocal z-stacks were acquired using an Olympus UPLSAPO × 40/NA 1.25 or × 30/NA 1.05 silicone oil immersion objectives. To determine the ISV diameter, a line was drawn perpendicularly across the ISV in Fiji (NIH). The intensity profile along this line was used to identify two peaks corresponding to the vessel walls. A horizontal line was manually drawn at approximately half the maximum intensity of the peaks to measure the diameter. Over three measurements were averaged for each ISV (No.5-18). For overexpression experiments, vessel diameters were measured specifically in regions containing GFP/mCherry-positive overexpressed cells. These regions were often part of mosaic vessels comprising both OE and non-OE cells. Control measurements were taken from either non-mosaic vessels or mosaic vessels lacking GFP/mCherry expression. The length of each ISV was measured by manually tracing a polyline along the vessel's spine in Fiji. The tracing began at the point where the ISV connected to the DLAV and extended down to its connection with the DA or PCV. To count ECs, the nuclei within ISVs No.5-18 were manually counted in Fiji.

## Analysis of EC exchange, division, and rearrangement in timelapses

Confocal z-stacks were acquired using an Olympus UPLSAPO × 40/NA 1.25 silicone oil immersion objectives. Timelapse imaging was performed in *Tg(fli1a:H2B-EGFP)*[ncv69];*Tg(fli1:Lifeact-mCherry)*[ncv7] or *Tg(fli1:GAL4FF)*[ubs3]; *Tg(UAS:EGFP-UCHD)*[ubs18] embryos from 2-3 dpf and 3-4 dpf. Endothelial cell exchange was tracked by following nuclear movements in 2D maximum-projected time-lapses. To visualise cell rearrangements, imaged vessels were straightened and divided into front and back views, enabling the tracking of individual ECs based on junctional actin. Cell division was identified by the transient accumulation of F-actin and the subsequent appearance of daughter nuclei in the time-lapse series.

## In vivo cell shape and strain analysis

Single-cell labelling of ECs within ISVs was achieved through mosaic expression of different constructs: *fli1ep:lynEGFP* plasmid in wild-type embryos, *6xUAS:myl9bA2A3* for overexpression experiments, or transplanted *krit1* mutant cells in *Tg(fli1:GAL4FF)*[ubs3], *Tg(UAS:EGFP-UCHD)*[ubs18] transgenic background. At 2, 3, and 4 dpf, microangiography was performed, and vessels were imaged using an Olympus UPLSAPO 60 ×/NA 1.2 water immersion objective with an optical z-plane interval of 0.25 μm. EC shape analysis was conducted

semi-automatically using a custom-written Fiji script developed in Python, adapting methods from a previously described approach[12]. The analysis provided measures of cell area (a) and aspect ratio (r), which were subsequently utilised for strain analysis. Cell at radial axis (cell width, **w**) and axial axis (cell length, **l**) were calculated as $\mathbf{w} = \sqrt{\frac{a}{r}}$; $\mathbf{l} = \sqrt{ra}$. Effective cell number was calculated as $N_{eff}^{(radial)} = \frac{\pi D}{w}$ at the radial axis, and $N_{eff}^{(axial)} = \frac{L}{l}$ at the axial axis, where **D** is vessel diameter, and **L** is vessel length. Cell number strain was calculated from the effective cell number ($N_{eff}$). For a quantity **X** (e.g., vessel diameter **D**, vessel length **L**, cell width **w**, cell length **l**, effective cell number $N_{eff}$), strain (ε) between two time points (**t**) was calculated as:

$$\varepsilon_X(t_1 \rightarrow t_2) = \frac{X_2 - X_1}{X_1}$$

For example, vessel strain at the radial axis was calculated from vessel diameter, as:

$$\varepsilon_D(t_1 \rightarrow t_2) = \frac{D_2 - D_1}{D_1}$$

and analogous formulations were used for axial vessel strain (from vessel length), radial and axial cell strain (from cell width and cell length), and radial and axial cell-number strain (from $N_{eff}^{(radial)}$ and $N_{eff}^{(axial)}$).

### Analysis of actin organisation

Processed images were analysed by dividing each ISV into front and back views. Since lateral mounting causes the dorsal side to face the objective while the ventral side is obscured by the yolk, only the dorsal portion of ISVs (at positions 10 to 18) was evaluated, where fine actin structures could be reliably resolved. Actin organisation was classified manually as circumferential (C), mesh-like (M), or longitudinal (L). Both dorsal views (front and back) of an ISV were examined; if patterns differed (e.g., front = M, back = C), the ISV was assigned a combined category (CM). Percentages were calculated as the number of ISVs in each category divided by the total number of analysable ISVs across embryos. These values therefore represent the fraction of ISVs exhibiting a given AO, not the proportion of actin structures within a vessel.

### Analysis of actomyosin colocalisation and dynamics

Processed images were analysed by dividing each ISV into front and back views. Multichannel z-stacks were maximally projected to 2D, and a region of interest (ROI) covering actin-myosin II colocalisation was manually drawn. For time-lapse experiments, intensity profiles for actin and myosin II were extracted over time in ImageJ, background-subtracted using the mean of a rectangular region, and converted to log10 values. To compare actin and myosin II on the same scale, $\log_{10}$ values across time for each vessel were further normalised by max-min scaling. Vessel diameter was measured from kymographs by tracing vessel edges and plotted together with normalised actin and myosin II intensities to assess temporal correlations. For static images of acto-myosin colocalisation in wild-type and *krit1* mutant embryos (obtained from in-crosses of heterozygous parents, resulting in homozygous embryos lacking blood circulation), 6 regions were manually drawn along junctions and 10 along cortical actin per vessel. Junctional actin was measured at cell–cell contacts, corresponding to the bright, continuous EGFP-UCHD signal. Cortical actin was measured within the cell interior but not contiguous with junctional pixels, usually with a lower EGFP-UCHD signal. Because endothelial cell membrane in vivo is extremely thin (~1 μm), and z resolution is limited, apical and basal membranes cannot be reliably distinguished; thus, the cortical region integrates both surfaces. Actin and myosin II intensities were measured in both channels, background-subtracted using the mean of two

regions outside the vessel, and averaged to yield mean junctional and cortical values. Ratios of junctional to cortical intensity were then calculated for actin and myosin II, converted to $\log_{10}$ values, and used to generate one averaged ratio per vessel for plotting.

### Laser ablation and analysis of mean velocity of recoiled actin

Laser ablations were performed using a Zeiss LSM980 confocal microscope equipped with adjustable laser line MP-NDD (69–1300 nm) and a 63x Oil immersion objective (Plan-Apochromat, NA 1.2). Ablation parameters were controlled via ZEN (Blue 3.8) software. Each ablation consisted of a single laser pulse at 920 nm with 80% laser power. Ablation was applied at the 5th frame of a time-lapse sequence comprising 50 frames captured at intervals of 250–300 ms (scan speed 13, and pixel time 0.75 μs), depending on the region of interest size. The location of ablation was manually drawn with a 0.3 μm × 6 μm box.

To quantify actin recoil velocity, the ablated region (a gap) in the actin network was analysed by drawing a line region of interest (ROI) across the gap to generate a kymograph. The edges of the gap were manually traced on the kymograph, and the distance between these edges was measured for each time frame to track the gap distance over time. Velocity was calculated by dividing the change in distance by the frame rate. The average velocity was calculated from measurements taken during the first 10 s of the time lapse. For each ablation, three separate line ROIs were drawn across the gap, generating three average velocity measurements. The final mean velocity for each ISV was determined by averaging these three values. Each data point in the velocity bar chart represents the mean recoil velocity of an individual ablated ISV.

### Model of cytoskeletal dynamics in endothelial cells in 2D

Since ECs lining blood vessels are flat, the system was approximated by a 2D model in a rectangular area of 8 μm (horizontal, circumferential) × 30 μm (vertical, longitudinal). The cytoskeletal network comprised filaments, motors, crosslinkers, and cell membranes, incorporating processes such as filament polymerisation/depolymerisation, motor-generated active forces, and turnover of filaments and crosslinkers to preserve network dynamics. To promote bundle formation, anchored filaments were introduced at the vertical (longitudinal) boundary or along the vertical midline. The simulation was first performed in a fixed boundary to see the bundle formation. Deformable membrane was then introduced along the vertical boundary to see the cell deformation induced by the bundle formation. This model represents an extension of our previously described framework[54], and full details are provided in the Supplementary Note 1 and Supplementary Table 1.

### Stress calculation

The stress tensor in the simulation was computed according to the following formula:

$$\sigma_{\alpha\beta} = \frac{1}{A} \sum_{i > j} \left\langle f_{ij,\alpha} r_{ij,\beta} \right\rangle,$$

where $A$ is the area of the system, $\mathbf{f}_{ij}$ is the force acting on bead $j$ by bead $i$, $\mathbf{r}_{ij} = \mathbf{r}_i - \mathbf{r}_j$, $f_{ij,\alpha}$ is the $\alpha$ component of $\mathbf{f}_{ij}$ and $r_{ij,\beta}$ is the $\beta$ component of $\mathbf{r}_{ij}$.

### Quantification of filament orientation

The degree of filament orientation in the circumferential direction (x direction) was quantified by circumferential alignment

$$N_x^2 = \sum_{i \in R_i} n_{i,x}^2$$

where $n_{i,x} = \cos\theta_i$ is the x component of the orientation $\theta_i$ of segment $i$. Here, each filament consists of multiple segments (See supplemental Method). This quantity was measured in the blue region in Fig. 5d in both peripheral and central (red region, between 2 and $5\,\mu m$) regions.

## Statistics and reproducibility

All statistical analyses were performed using GraphPad Prism 10. The specific tests used for each dataset are indicated in the corresponding figure legends. Comparisons between two groups were evaluated using unpaired two-tailed Student's *t* tests, while comparisons among more than two groups were assessed using one-way ANOVA followed by Tukey's post hoc test. Categorical data were analysed with Fisher's exact test. Exact *p*-values are reported in the figures. Statistical significance was defined as $p < 0.05$. Additional details are provided in the figure legends. The sample size (number of cells or vessels) was chosen based on the number of embryos expressing the transgene of interest (e.g., transient overexpression of plasmids) or of the desired genotype obtained per experiment. The number of embryos used for time-lapse imaging per experiment ranged from 3 to 4 per experiment for overnight imaging, 6 to 8 for short (e.g., 20−30 min) but high temporal resolution imaging, since time was a limiting factor. Data exclusions: (1) Cell area measurements: endothelial cells located in the ventral region of ISVs (connected to the dorsal aorta) were excluded as these cells are significantly smaller that cells located in the middle and dorsal regions of the ISVs. In addition, analysis of actin organisation was only performed in middle-dorsally located cells. 2) Analysis of actin organisation: ISVs with weak reporter expression, poor image quality, or undergoing remodelling without flow were excluded. Images or schematics in figures were organised or manually drawn using Adobe Illustrator 2022.

## Data availability

All data are available in the supplementary files. Source data are provided in this paper.

## Code availability

Custom code for in vivo cell shape analysis can be found here: https://github.com/dougkelly88/vessel_cell_shape_analysis. Other custom Fiji Macros used in this paper can be found here: https://github.com/yanc0913/CircActin (Zenodo https://doi.org/10.5281/zenodo.17731322)[55].

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

## Acknowledgements

We thank members of the Phng Lab for discussions and suggestions; Emi Taniguchi, RIKEN BDR Research Aquarium and RIKEN Kobe Biol-maging Facilities & Factory for technical assistance; Hironobu Fujiwara for access to the ZEISS LSM980 microscope; Holger Gerhardt, Salim Seyfried and Anne Schmidt for plasmids; Heinz-Georg Belting and Markus Affolter for providing the transgenic line *Tg(fli1:GAL4FF)*[ubs3], *Tg(UAS:EGFP-UCHD)*[ubs18]; Fumio Motegi for comments on the manuscript; and Yichen He for advice on Fiji macros and Python codes. This work was supported by core funding from RIKEN BDR (to L.-K.P.); RIKEN BDR DECODE project (to N.T.); RIKEN BDR STPJ Project (to L.-K.P.); RIKEN BDR-Otsuka Pharmaceutical Collaboration Centre (to Y.C.); RIKEN Junior Research Associate Programme (to M.H.); the JSPS Grants-in-Aid for Scientific Research grants (22H022624 and 22H05168 to L.-K.P; 22H05170 to S.O.; JP19H03394, JP19H05794, JP19H05795, JP22H02798, JP22H04926 to Y.O.; JST CREST grant JPMJCR1852 to Y.O.; JST Moonshot R&D grant JPMJMS2025-14 to Y.O.; AMED CREST grant JP23gm1700001s502 to Y.O.; ARC Discovery Project grant (DP230100393) to A.K.L.; NHMRC Ideas Grant (2029372) to A.K.L.; ARC (FL230100100) and DFG (553948485) to J.E; and LeDucq Transatlantic Network of Excellence "ReVAMP" (to L.-K.P.).

## Author contributions

L.K.P., Y.C., and N.T. conceptualised the project. Y.C., L.K.P., J.D.S., N.T., N.A., G.C., M.H., Y.O., and A.K.L. performed experiments. V.S., B.B., and T.S. conducted computational simulations, analyses, and prepared the corresponding figures and text. J.E. performed order parameter analysis of actin filaments. Y.C., N.T., L.K.P., I.K., M.H., N.A., G.C., and S.O. analysed all other data. Y.C. and N.T. developed Fiji macros for image processing. Y.C. prepared the figures and movies. Y.C. and L.K.P. wrote and edited the manuscript. L.K.P. and A.K.L. acquired funding, and L.K.P. supervised the project. All authors discussed the results and commented on the manuscript.

## Competing interests

The authors declare no competing interests.
