## [Transparent Peer Review file · Nature Communications]

Circumferential actomyosin bundles anchored by CCM1 drive endothelial cell contraction and vessel constriction

Corresponding Author: Dr Li-Kun Phng

Version 0:

Reviewer comments:

Reviewer #1

(Remarks to the Author)

This is a nicely written paper in which the authors use transgenic zebrafish lines and confocal imaging to measure vessel and endothelial cell morphometrics in arterial (a) and venous (v) intersegmental vessels (ISVs) from 2-4 days post-fertilization. They find that aISVs contain fewer, larger ECs and vISVs contain more, smaller ECs, with increased EC number arising largely from cell divisions after sprouting and anastomosis are complete. These findings confirm and modestly extend (to later timepoints) previously published work (e.g. Rosa et al. Development 2022). Beyond these morphometric analyses, the authors focus on the role of different types of actin in controlling EC size and vessel diameters. They suggest that circumferential actomyosin is the key driver of constriction in aISVs and vISVs between 2-4 dpf. While this is a logical idea, more rigor is required to establish causation or strong correlation. The imaging is well-presented and morphometric measurements seem sound. The work is mostly descriptive and provides modest new insight into the mechanism of endothelial cell shape changes and vessel diameter changes.

Major concerns

1. The primary novel conclusion from this work is that circumferential actomyosin is responsible for EC contraction and decreased ISV radius between 2 and 4 days post-fertilization (dpf). However, data are largely observational, and temporal changes in actin organization and cell size and vessel diameter do not always correlate. For example, the authors do not address how circumferential actin, present primarily at and analyzed only at 2 dpf, accounts for diameter decreases in aISV and vISV, which are only measurable at 4 dpf (Fig. 1b, and presumably Fig. 5e,f; 6g,h, although stats not shown). How is tension maintained and diameter restricted in the absence of these structures at 4 dpf?
2. Similarly, at 2 dpf, even on short time scales, the apparent loss of circumferential actin over time correlates with constriction and maintenance of smaller vessel diameter (Fig. 3e,f). Can the authors show, in wild type ISVs, that regions that have circumferential actin contract more than regions without it, on this same time scale?
3. Actomyosin/diameter correlation (Fig. 4a-c) was assayed only at a region of circumferential actomyosin. Additional correlations between actin/myosin/diameter should be shown, both for regions with high and low colocalization, to strengthen conclusions.
4. In Fig. 5, while there is an apparent decrease in pure circumferential actin at 2 dpf in aISV and vISV with wasb-OE, as expected, there is an apparent increase in mixed (mesh + circumferential) actin with wasb-OE at 3 dpf in aISV and vISV, and at 4 dpf in aISV. Accordingly, these data do not strongly support a role for circumferential actin in constriction.
5. The authors do not directly consider a role for pericytes in narrowing of ISV diameter, Pericytes are present on ISVs as early as 2 dpf (Ando Development 2016; ref. 29; Colijn et al. Dev Dyn 2024).
6. The percent of each actin type is not consistent in control in Figs. 3, 5, 6, suggesting high variability from clutch to clutch, inconsistent timing, and/or high interoperator variability/subjectivity from observer to observer. Can the authors please address this?
7. In Fig. 6k,l, myosin disruption seems to increase aEC size and decrease radial strain, diameter. By contrast, the same treatment increases vEC size but has no effect on radial strain. Why are these parameters uncoupled in vISVs and coupled in aISVs?
8. The authors nicely detail morphometric differences in arterial versus venous ISVs, but do not explore the mechanisms that underlie these differences. Do the authors think that the differences simply reflect increased proliferative potential of vECs?

Minor concerns

1. All movies would benefit from annotation to draw the eye to events of interest.

2. Figure 1b shows a significant increase in aISV diameter and non-significant decrease in vISV between 2 and 3 dpf, but this is buried by a general statement that “both vessel types exhibit a significant reduction in diameter...” over this time (2-4 dpf).
3. The methods state that ISVs 10-18 (so 9 per embryo) were analyzed, but figures and legends say that N = total number of ISVs. So for 3b, for example, n= 42 could mean a few as $42/9 = 4-5$ embryos. For statistical purposes, intra-embryo measurements should be averaged, and each N should be a single embryo, not a single ISV. Regardless, number of embryos analyzed should be clearly stated.
4. Is there any correlation between actin type and position within ISV? For example, the circumferential actin shown in Figure 3a is at a bend.
5. Line 240: vISVs (9 ECs at 3 dpf) are longer than aISVs (5 ECs at 3 dpf), so cell numbers could be normalized to length for fairer comparisons.
6. In the myl9bA2A3-EGFP overexpression experiments, no gal4 driver is mentioned in the text, Fig. 5 legend, or methods. The only other transgene described is Tg(fli1:myr-mCherry).
7. Figure 4a-c: Data are from an aISV at 2 dpf. Were the additional data (from total of 20 movies, not shown) all aISV, or also vISV?
8. Figure 5b, c: statistical analysis required to prove that wasb-OE alters actin organization
9. Line 316-317, Fig. 5g-i: Results statement needs to be clarified. There is no increase in diameter in vISV with myl9bA2A3-OE at 3 dpf. Also, the authors should clarify whether vessel diameters were measured in mosaic WT and OE ISVs (as shown in g), non-mosaic vessels, or a mixture of both. Particularly interesting would be to measure diameters in WT and OE regions of mosaic vessel.
10. Line 319, Fig. S4: The only significant cell area increases with myl9bA2A3-OE are at 4 dpf in both aISV and vISV; it should be stated in the text that there are no increases at 2 and 3 dpf.
11. Line 360, Fig. S5: Text says the area of *krit1*^{-/-} ECs is increased, but this is not true for aECs at 2 dpf and not significant for vECs at 2 dpf.
12. Line 362, Fig 6g,h: Text states that both ISV types have “significantly enlarged lumens at 2, 3, and 4 dpf compared to wild-type and *krit1*^{+/-} EC-containing vessels.” This is untrue according to statistical analysis.
13. The DA and PCV also get smaller in diameter over early developmental time. Can you speculate on whether your findings are likely applicable to larger multicellular tubes?
14. Reference 15 is now published, please update.

Reviewer #2

(Remarks to the Author)

This manuscript by Chen et al. addresses the mechanisms that lead to zebrafish intersegmental vessels remodeling during development. Whether and how acto-myosin contraction allows vessel constriction during vessel remodeling is the driving question. A recent paper in *Angiogenesis* by Yin et al. 2024, has investigated the role of acto-myosin contraction at cell-cell junctions for lumen formation in zebrafish without addressing the question of vessel caliber. Therefore, the questions here have yet not been addressed directly.

However, the general claim that circumferential actomyosin bundles and not the other actin organizations drive endothelial cell deformations is not enough demonstrated. Moreover, the contribution of other described phenomena such as cell exchange and cell division to vessel constriction is not well taken into account.

Major comments:

- 1- First, it is difficult to follow the terminology describing the different events. The exact meaning of terms like cell rearrangement, cell exchange, cell shape deformation, cell size and their relationship is not clear and makes figure 1 difficult to follow. Terms should be more clearly explained and maintained consistent between methods and the main results section. The results discuss the tissue strain while the methods describe the calculations for a vessel strain which can be confusing.
- 2- The method description needs to be much more clear and a schematic should be added to figure 1 to explain how the geometry of the cell is measured with an ellipse depicted with the major and minor axis which then leads to the various calculations described. Cv and Cr should be more clearly described as the length of the radial and vertical axes of the ECs assuming an ellipse.
- 3- In figure 2b, it is quite difficult to follow the variation in diameter of the vessels at the front and back sides. A curve representing actin intensity and vessel diameter over time as in figure 4c would help. The respective contribution of actin rings and cell rearrangement (with new cells entering the zone) is not clear.
- 4- The following part of the manuscript describes the role of actin remodeling on vessel constriction. However, this relies on a manual annotation of three types of actin which is a highly biased method. For example in fig3c, d, fig 5, fig6, the manual annotation of the actin fibers does not seem very close to the reality. Actin segmentation and categorization should be automated to make it more reliable. In Fig 3, the percentage of the various actin organizations is quantified. But then the spatiotemporal relationship between actin organization and vessel/cell shape changes is only described for circumferential actin. Are the other actin types also involved ? if not, this needs to be shown to claim that it is primarily circumferential actin that drives vessel constriction.
- 5- This brings up the numbers of technical compared to biological replicates which are not clear. Indeed, there is no clear

reporting throughout the entire paper of independent trials/embryos relative to the vessels or cells/vessel quantified. The “n” is reported above several bar plots but very unclear what this n includes. Moreover, there is no appropriate statistical reporting section in the methods where this should also be elaborated upon. On first glance, since source data files are missing, the p values look very low for the distribution that can be seen in the bar plots.

6- Lines 308-309 conclude that circumferential actin is essential for vessel constriction but the OE-vessels still constrict and the difference between control and OE is 17 versus 14% which seems hardly enough to support this conclusion.

7- One interesting point raised is the contribution of flow to vessel dilation in Krit^{-/-} embryos. This still open question in the field needs more investigation on how no flow exacerbates CCM phenotype, and in what ways – actin/cell shape/proliferation/migration. Therefore, the effect of flow on the Krit phenotype should be more thoroughly explored and quantified. A movie of krit1^{-/-} ISV with flow or no flow should be shown.

Minor comments:

8- Fig. 4d: which dpf is this comparison performed on? For vISVs this difference between recoil velocity of circumferential and longitudinal actin is really not significant and the sample size is also small. Are the vISVs from different embryos or within one independent trial?

9- Lines 302-305- claim that there is a reduction in circumferential actin in both aISVs and vISVs at 2/3/4dpf in wasb OE ECs. This seems to not correlate with the box plots (eg. 3 and 4dpf for aISVs show higher C-M in the wasb-OE).

10- Lines 313-331: the section discusses the effects on cell and vessel strains from Myosin II inhibition but does not discuss the effects on cell number changes which are identified as an important factor in Fig. 1. A quantification of ES/number of cells could help clarify this.

Reviewer #3

(Remarks to the Author)

Summary of the study: Using the zebrafish model, the authors intent to address how transitions in endothelial cell (EC) number and shape are coordinated to define vessel diameter during development remains. Their excellent data revealed that EC deformations, rearrangements and transient formation of self-seam junctions play key roles in regulating cell number and vessel diameter. By performing high-resolution analysis of actin cytoskeleton organization disclosed the generation of tension bearing, circumferential actomyosin bundles in the endothelial cortex that drive EC deformation and vessel constriction. In a vascular disease model of CCM, they showed that the loss of circumferential actin bundles in krit1/ccm1-deficient ECs causes cell enlargement and impaired vessel constriction that culminate in dilated vessels. Their strong data support that circumferential actomyosin-driven EC deformations in controlling vessel size and in the prevention of vascular malformations.

This is a very novel and well performed study with high quality data. I only have a minor issue regarding the myosin II activity in CCM disease model. They showed inhibition of myosin II activity leads to disrupted cell deformation and increased vessel diameter. Yet, CCM samples exhibited increased pospho-MLC2 thus increased myosin activity. How these can be reconciled?

Reviewer #4

(Remarks to the Author)

Version 1:

Reviewer comments:

Reviewer #1

(Remarks to the Author)

The authors have very nicely addressed all of my previous concerns regarding the original submission, as evidenced by numerous additional experiments presented in both the main text and in the supplement. They convincingly demonstrate a role for circumferential actomyosin in endothelial cell contraction and vessel narrowing and new data strengthen support for a role for the CCM protein, KRIT1, in this process.

Reviewer #2

(Remarks to the Author)

First we would like to acknowledge the authors for their efforts in thoroughly revising the manuscript. The use of a consistent terminology over the manuscript helps its understanding like the new explanations on the quantified parameters.

Additional data have been acquired for many of the figures improving the robustness of the statistical analyses and numbers of studied vessels, embryos and experiments have been specified as requested.

New experiments with overexpression of Krit and Fascin nicely complement the data.

However, some parts still need attention:

-We noticed that in Fig1c only one experiment has been performed to quantify EC number per vessel at 4 dpf which is surprising as many other quantifications have been done at this time point in other figures.

-In figure 2g, at 3-4 dpf, it is said that aEC shrinks while vEC shortens. Please explain what it means either directly in the figure or in its legend.

-In figure 4c-f, it is not said whether these are aISV or vISV.

-In the paragraph line 298-313, it should be mentioned that whereas circumferential actin is more enriched at 2 dpf, diameter constriction is only statistically significant at 3 and 4 dpf.

-Line 377: please cite the original paper from Glading et al., JCB 2007.

- Paragraph 422-431 and sup fig 7h: a new Junction/Cortex Ratio is introduced in the quantifications but the term cortex is not clearly defined. Does this "cortex actin" actually correspond to the total intracellular actin content or to only a zone and which one ?

Concerning the linear fibers that are present in krit1^{-/-} ECs (Fig 7g), why are they not considered as circumferential actin fibers ? They also cross the cell from one junction to another. We are not necessarily asking for more experiments but for more clarity about what is seen here and its interpretation. Obviously, the wrong location in z of these contractile fibers is key for the krit^{-/-} phenotype and could explain the paradox raised by reviewer 3 concerning the increased pMLC in these cells.

-Line 484: in Lisowska et al (ref 29), it is shown that in 2D, ROCK1 promotes ventral actin stress fibers connected to focal adhesions and ECM and not "cortical stress fibers" as said here.

-Line 491: Please refer to fig 7e when talking about transplanted cells exposed to flow, it will facilitate the comparison. In this paragraph, mosaicism should also be commented as it may also impact the acto-myosin organization of the mutant cell and may contribute to the differences between pure homozygous vessels and transplanted populations.

-Line 539: When talking about the effect of CCM mutant cells on neighbouring ones, Shapeti et al, Nat com 2024 should also be cited.

-Movies 8 and 18 don't work.

Reviewer #3

(Remarks to the Author)

The authors have carefully addressed my previous concerns on Rock/MLC activation in CCM pathogenesis.

Reviewer #4

(Remarks to the Author)

Response to Reviewers' Comments

We thank the reviewers for their valuable feedback. Below, we provide point-by-point responses (in blue) to all comments and have made corresponding revisions to address these concerns.

Reviewers' Comments:

Reviewer #1 (Remarks to the Author)

This is a nicely written paper in which the authors use transgenic zebrafish lines and confocal imaging to measure vessel and endothelial cell morphometrics in arterial (a) and venous (v) intersegmental vessels (ISVs) from 2-4 days post-fertilization. They find that aISVs contain fewer, larger ECs and vISVs contain more, smaller ECs, with increased EC number arising largely from cell divisions after sprouting and anastomosis are complete. These findings confirm and modestly extend (to later timepoints) previously published work (e.g. Rosa et al. Development 2022). Beyond these morphometric analyses, the authors focus on the role of different types of actin in controlling EC size and vessel diameters. They suggest that circumferential actomyosin is the key driver of constriction in aISVs and vISVs between 2-4 dpf. While this is a logical idea, more rigor is required to establish causation or strong correlation. The imaging is well-presented and morphometric measurements seem sound. The work is mostly descriptive and provides modest new insight into the mechanism of endothelial cell shape changes and vessel diameter changes.

Major concerns:

1. The primary novel conclusion from this work is that circumferential actomyosin is responsible for EC contraction and decreased ISV radius between 2 and 4 days post-fertilization (dpf). However, data are largely observational, and temporal changes in actin organization and cell size and vessel diameter do not always correlate. For example, the authors do not address how circumferential actin, present primarily at and analyzed only at 2 dpf, accounts for diameter decreases in aISV and vISV, which are only measurable at 4 dpf (Fig. 1b, and presumably Fig. 5e,f; 6g,h, although stats not shown).

We thank the reviewer for pointing out the discrepancy between the stage of circumferential actin analysis and the stage at which vessels become significantly narrower. We revisited the data that we had used for the first submission and realized that each data point represented one diameter measure, not the average diameter per vessel. We apologize for this mistake and have corrected it so that each data point now represents the average of a vessel. Additionally, we have included average diameter measurements from at least two additional independent experiments. With these corrections and updates, we observe that

diameter decreases from 2 dpf for aISVs (Fig. 1b), when circumferential actin is most frequent. This continuous vessel narrowing from 2 dpf is also observed in control or wildtype vessels in Fig. 6d and f, Supplementary Fig.6d and Supplementary Fig.7b. Time-lapse imaging provide further evidence of diameter fluctuations with transient narrowing events at 2 dpf (Fig. 4E, Supplementary Movie 7), demonstrating that vessel constriction occurs as early as 2 dpf (Fig. 1b). Together, these data demonstrate that EC constriction begins at 2 dpf, but significant reduction in vessel diameter becomes apparent by 4 dpf.

We also repeated the strain analyses at two separate intervals, 2-3 dpf and 3-4 dpf, instead of a single 2-4 dpf interval, to gain a more detailed understanding of endothelial cell (EC) behaviours during vessel remodelling (Fig. 2c – g, lines 135 - 162). Specifically, we calculated (i) **cell number strain**, indicating how many cells span the vessel axis; (ii) **cell strain**, reflecting EC shrinkage or expansion; and (iii) **vessel strain**, representing the net tissue-level change. These parameters clarify how cell rearrangement and size changes contribute to vessel constriction. Strain analysis revealed that aISV **constriction begins at 2 dpf**, when cell addition coincides with EC narrowing, and is later reinforced by axial rearrangement. In vISVs, early proliferation increases cell number but is offset by EC narrowing, with later constriction driven primarily by axial cell rearrangement. Together, these findings show that **circumferential actin enrichment at 2 dpf marks the onset of endothelial contraction and vessel constriction**, while the measurable diameter reduction by 4 dpf reflects the cumulative outcome of these early contractile events.

How is tension maintained and diameter restricted in the absence of these structures at 4 dpf?

Laser ablation experiments revealed tension in circumferential, mesh, and longitudinal actin, with circumferential actin generating more than 1.5 times higher tension than mesh and longitudinal actin (Fig. 5f). This elevated contractile force in circumferential actin is essential for cell deformation at 2-3 dpf, driving vessels toward their final diameter. Following this initial cell contraction phase, mesh and longitudinal actin organizations likely function to maintain the reduced cell size. This concept aligns with our previous findings (Kondrychyn et al., 2020), which demonstrated that decreased branched actin results in enlarged ECs and consequently dilated vessels. We therefore propose that while circumferential actin bundles provide the primary contractile force for cell deformation, mesh and longitudinal actin bundles contribute to the cortical tension necessary to stabilize the contracted cell and constricted vessel. Although cell-cell junctions were not investigated during vessel remodelling in this study, junctional tension likely exists (Supplementary Fig. 7g and h, *krit1^{+/+}*) and contributes to cell size and vessel shape regulation. The coordinated action of these distinct actin networks—circumferential bundles for active contraction and mesh/longitudinal networks for structural maintenance—ensures precise control of vessel diameter during development.

2. Similarly, at 2 dpf, even on short time scales, the apparent loss of circumferential actin over time correlates with constriction and maintenance of smaller vessel

diameter (Fig. 3e,f). Can the authors show, in wild type ISVs, that regions that have circumferential actin contract more than regions without it, on this same time scale?

We have plotted the diameter of vessels exhibiting circumferential, mesh and longitudinal actin on the same time scale in Supplementary Fig. 4. These additional analyses show that in 11 of 12 movies, the formation of circumferential actin correlates with decrease in vessel diameter. In 8/12 cases, the dissipation of circumferential actin is followed by vessel widening. In contrast, the formation of mesh actin formation is correlated with vessel widening (5 of 6 movies). Longitudinal actin formation is associate with varying degree of vessel constriction. This is mentioned in lines 217 - 224.

3. Actomyosin/diameter correlation (Fig. 4a-c) was assayed only at a region of circumferential actomyosin. Additional correlations between actin/myosin/diameter should be shown, both for regions with high and low colocalization, to strengthen conclusions.

We have plotted the correlation between myl9b colocalization at circumferential actin and diameter in Supplementary Fig. 5b. In the 16 vessels analyzed, periods of low actomyosin colocalization is consistently correlated with wider diameter while periods of high colocalization (grey box) is correlated with narrowing diameter. These observations are mentioned in lines 250 - 252.

4. In Fig. 5, while there is an apparent decrease in pure circumferential actin at 2 dpf in aISV and vISV with wasb-OE, as expected, there is an apparent increase in mixed (mesh + circumferential) actin with wasb-OE at 3 dpf in aISV and vISV, and at 4 dpf in aISV. Accordingly, these data do not strongly support a role for circumferential actin in constriction.

We agree with the reviewer that the wasb-OE experiment weakly supports the role of circumferential actin in vessel constriction and have moved these data to Supplementary Fig. 6a - d. We have instead worked towards strengthening the causal role of circumferential actin in vessel constriction by performing the following:

- i) Endothelial overexpression of Fascin 1, an actin bundling protein (Fig. 6a -d, lines 302 - 313). These new data show that Fascin 1-OE increased the number of vessels with circumferential actin formation and augmented the constriction of aISVs and vISVs.
- ii) Endothelial overexpression of Krit1 (Fig. 7b-d, lines 380 - 386). Elevated Krit1 expression also resulted in more vessels with circumferential actin and enhanced vessel constriction.
- iii) Mathematical modelling of F-actin, myosin and crosslinkers enclosed by a membrane revealed that the stress generated in the circumferential direction was larger than that in the longitudinal direction, and that the generated stress can deform the membrane (Fig. 5d-e, lines 254 - 284).

5. The authors do not directly consider a role for pericytes in narrowing of ISV diameter, Pericytes are present on ISVs as early at 2 dpf (Ando Development 2016; ref. 29; Colijn et al. Dev Dyn 2024).

Thank you for raising the question of whether pericytes have a role in ISV constriction. Although pericytes are present on ISVs as early as 2 dpf, they do not completely cover ISVs such that by 120 hpf (5 dpf), 40% of ISVs remained uncovered by pericytes (Ando et al., 2016, Development). Furthermore, there is preferential coverage of aISVs than vISVs, with only ~50% covered by pericytes (Ando et al., 2016, Development). Given the scarce coverage, especially around vISVs, we hypothesized that pericytes do not play a significant role in developmental vessel constriction. Nevertheless, we addressed Reviewer 1's concern by examining *pdgfrb*^{-/-} zebrafish, which lack functional *Pdgfrβ* signalling and have decreased number of mural cells around ISVs and cerebral central artery (Ando et al., 2016, Development). By measuring aISV and vISV diameters at 2, 3 and 4 dpf in *pdgfrb*^{+/+}, *pdgfrb*^{+/-} and *pdgfrb*^{-/-} zebrafish (Supplementary Fig. 9), we found that both aISVs and ISVs in *pdgfrb*^{-/-} zebrafish constrict to the same extent as wildtype embryos, with no statistical difference in diameter at each developmental stage. These results show that pericytes do not regulate ISV constriction during developmental vascular remodelling. This new data is discussed in **lines 503 - 507**.

6. The percent of each actin type is not consistent in control in Figs. 3, 5, 6, suggesting high variability from clutch to clutch, inconsistent timing, and/or high interoperator variability/subjectivity from observer to observer. Can the authors please address this?

The imaging sessions for 2, 3, and 4 dpf embryos were conducted within an approximate 4-hour window due to imaging throughput across several independent experiments. Although actin remodelling is dynamic as shown in our 1-minute time-lapse recordings, we also noticed that actin organizations oscillate and generally remain within the same classification for extended periods. To confirm stability, we re-examined embryos imaged first after completing the session and observed that their ISV actin patterns were comparable to those recorded ~4 hours earlier. Thus, the observed differences between developmental stages are unlikely to result from minor timing variation within each imaging batch.

The inconsistency in actin classification in control cells between experiments may be due to different numbers of ECs analyzed - fewer samples were analyzed in the Wasb-OE and Krit1 transplantation experiments compared to wildtype ECs. We have therefore performed more experiments to increase the number of control ECs for Wasb-OE experiment, Krit1^{+/+} ECs as well as WT ECs (see Table 1 below) and replotted the proportion of different actin organizations found in control ECs from the different experimental conditions (see figure below). By increasing the sample size, the proportion of actin type becomes more similar to wildtype, especially for vISVs at 2, 3 and 4 dpf and aISVs at 2dpf. In aISVs at 3 and 4 dpf, where sample size is small, there is still a slight disparity in actin type which will likely be corrected if sample size is increased even further. Regardless, we still observe a decrease in the

proportion of circumferential actin and an increase in longitudinal actin from 2 to 4 dpf for each control conditions.

Experiment	Vessel type	Developmental stage	No. of vessels (first submission)	No. of vessels (resubmitted)	No. of increased samples
Wildtype (Fig. 3b)	aISV	2 dpf	42	44	2
		3 dpf	46	62	16
		4 dpf	36	50	14
	vISV	2 dpf	19	56	37
		3 dpf	48	82	34
		4 dpf	26	77	51
Control ECs in Wasb OE experiment (Fig. 5b & c)	aISV	2 dpf	15	26	11
		3 dpf	10	19	9
		4 dpf	7	11	4
	vISV	2 dpf	11	16	5
		3 dpf	6	15	9
		4 dpf	7	15	8
krit1^{+/+} ECs (Fig. 6b & c)	aISV	2 dpf	11	20	9
		3 dpf	10	15	5
		4 dpf	10	11	1
	vISV	2 dpf	7	12	5
		3 dpf	13	18	5
		4 dpf	6	11	5

Another possible explanation for the difference in the proportion of actin classification is that the experimental condition to analyze actin organization in control ECs in the wasb-OE experiment is different from that used for analyzing actin organization in wildtype embryos. In the wasb-OE experiment, control ECs analyzed are wildtype ECs in embryos with mosaic endothelial overexpression of Wasb. In this experimental setting, Wasb-OE induces regional dilation of blood vessels (new Supplemental Fig. 6a, c and d) that can alter wall shear stress (WSS) distribution in the vascular network (Maung Ye and Phng, 2023). Surrounding wildtype ECs may respond to the altered WSS by remodelling its actin network. This experimental condition differs from the condition used to classify actin organization of wildtype ECs (Fig. 3b) where wildtype embryos were imaged, and their direct comparison should be treated with caution.

7. In Fig. 6k,l, myosin disruption seems to increase aEC size and decrease radial strain, diameter. By contrast, the same treatment increases vEC size but has no effect on radial strain. Why are these parameters uncoupled in vISVs and coupled in aISVs?

We appreciate the reviewer's point and have clarified the analysis by separating intervals (2-3 and 3-4 dpf) and explicitly reporting cell strain, vessel radial strain, and radial cell-number strain (formerly E.number). Importantly, size (absolute dimension) and strain (temporal change) describe distinct aspects of vessel remodelling.

In aISVs, myosin II inhibition prevented normal EC narrowing, showing positive cell strain at both intervals (Fig. 6h). Early constriction (2-3 dpf) was partly preserved by enhanced radial-to-axial rearrangement (negative cell number strain), but this compensation weakened by 3-4 dpf, leading to vessel widening. In vISVs, reduced EC narrowing combined with stronger radial-to-axial redistribution at 2-3 dpf produced modest constriction, while later cell expansion offset rearrangement, resulting in weaker constriction overall.

Although vISVs exhibited similar net vessel strain to controls, their ECs possess less circumferential actin and experience higher proliferative compaction, which may stabilize diameter despite altered contractility. Overall, myosin II inhibition disrupts the coordination between cell contraction and rearrangement, impairing vessel constriction.

8. The authors nicely detail morphometric differences in arterial versus venous ISVs, but do not explore the mechanisms that underlie these differences. Do the authors think that the differences simply reflect increased proliferative potential of vECs?

As venous ECs proliferate more frequently than arterial EC, ECs in the vISVs may be subjected to more compressive forces that may promote vEC shrinkage. We attempted to address this question by inhibiting cell proliferation with aphidicolin and hydroxyurea.

However, this treatment resulted in the reduction of endothelial cell number in both venous and arterial ISVs, preventing us to investigate the proliferative capacity of vECs on vessel morphometrics (Supplemental Fig. 10). This is discussed in lines 511 -517.

Minor concerns

1. All movies would benefit from annotation to draw the eye to events of interest.

All movies are now annotated.

2. Figure 1b shows a significant increase in aISV diameter and non-significant decrease in vISV between 2 and 3 dpf, but this is buried by a general statement that “both vessel types exhibit a significant reduction in diameter...” over this time (2-4 dpf).

We revisited and corrected our data, and added additional data as discussed above. The revised analysis shows that aISVs display a gradual, significant reduction in diameter between 2 and 4 dpf, whereas vISVs showed similar diameter between 2-3 dpf and constricted between 3-4 dpf (lines 96-98).

3. The methods state that ISVs 10-18 (so 9 per embryo) were analyzed, but figures and legends say that N = total number of ISVs. So for 3b, for example, n= 42 could mean a few as $42/9 = 4-5$ embryos.

This is not always the case as not all ECs are bright enough to image. ISVs with weak reporter expression, poor image quality, or undergoing remodelling without flow were excluded. The number of embryos and experiments are now included in the figure legend.

For statistical purposes, intra-embryo measurements should be averaged, and each N should be a single embryo, not a single ISV. Regardless, number of embryos analyzed should be clearly stated.

We have found that aISVs (or vISVs) do not all exhibit the same actin organisation in the same embryo. The reason for the heterogeneity in actin organisation is unclear but may represent different EC status/behaviour (cell rearrangement, proliferative) or a consequence of different perivascular environment (shear stress/pressure level, ECM deposition). Therefore, it is not possible to “average” actin organisation from one embryo.

We have now clearly stated that n = ISV, and the number of embryos analyzed and experiments performed are now stated in figure legends.

4. Is there any correlation between actin type and position within ISV? For example, the circumferential actin shown in Figure 3a is at a bend.

We thank the reviewer for this observation. Because the ventral region of ISVs lies deeper in the tissue and cannot be imaged at sufficient resolution, we restricted our analysis of

cortical actin organization to the dorsal part of ISVs, where imaging quality allows reliable classification (see Methods). At 2 dpf, 14% of aISVs displayed circumferential (C) and 34% C-M actin organizations. Within this dorsal region, circumferential actin was not restricted to vessel bends (in approximately 47% of ISV showing C/C-M patterns), while in the remaining vessels it occurred along straight segments. Thus, circumferential actin is not positionally confined but can arise throughout the dorsal ISV.

5. Line 240: vISVs (9 ECs at 3 dpf) are longer than aISVs (5 ECs at 3 dpf), so cell numbers could be normalized to length for fairer comparisons.

Upon the reviewer's suggestion, we have plotted EC number per 100 μm vessel and added this graph in Supplemental Fig. 1a. We have kept the graph showing the absolute number of ECs per aISV or vISV in Fig. 1c.

6. In the myl9bA2A3-EGFP overexpression experiments, no gal4 driver is mentioned in the text, Fig. 5 legend, or methods. The only other transgene described is Tg(fli1:myr-mCherry).

We have now included the gal4 driver line in the text (line 331).

7. Figure 4a-c: Data are from an aISV at 2 dpf. Were the additional data (from total of 20 movies, not shown) all aISV, or also vISV?

We have now shown the analysis in Supplementary Figure 5b and the vessel type (aISV or vISV) is written above each plot. These data are further mentioned in main text in lines 250-252.

8. Figure 5b, c: statistical analysis required to prove that wasb-OE alters actin organization

We have now performed Fisher's exact test to analyse categorical data. The shift in AO in wasb-OE vessels is mentioned in lines 320-322.

9. Line 316-317, Fig. 5g-i: Results statement needs to be clarified. There is no increase in diameter in vISV with myl9bA2A3-OE at 3 dpf. Also, the authors should clarify whether vessel diameters were measured in mosaic WT and OE ISVs (as shown in g), non-mosaic vessels, or a mixture of both. Particularly interesting would be to measure diameters in WT and OE regions of mosaic vessel.

We thank the reviewer for this comment and have clarified the text in lines 333-335. Vessel diameters were measured specifically in regions containing GFP-positive Myl9bA2A3-OE cells. These regions were often part of mosaic vessels comprising both OE and non-OE cells. Control measurements were taken from either non-mosaic vessels or mosaic vessels lacking GFP expression. This is mentioned in Methods in lines 643-647.

10. Line 319, Fig. S4: The only significant cell area increases with *myl9bA2A3*-OE are at 4 dpf in both aISV and vISV; it should be stated in the text that there are no increases at 2 and 3 dpf.

This is now clarified in lines 337-339.

11. Line 360, Fig. S5: Text says the area of *krit1*^{-/-} ECs is increased, but this is not true for aECs at 2 dpf and not significant for vECs at 2 dpf.

We added more experiments (see table below) and have corrected the text to state that “Cell area in homozygous mutants remained similar to controls at 2 dpf but increased significantly at 3 dpf in vECs and in both vessel types by 4 dpf”. Lines

Genotype	Vessel type	Developmental stage	No. of cells (first submission)	No. of cells (resubmitted)	No. of increased samples
Wild type	aISV	2 dpf	4	12	8
		3 dpf	11	14	3
		4 dpf	13	17	4
	vISV	2 dpf	5	7	2
		3 dpf	6	12	6
		4 dpf	8	14	6
Heterozygous	aISV	2 dpf	17	23	6
		3 dpf	12	18	6
		4 dpf	14	14	0
	vISV	2 dpf	10	25	15
		3 dpf	14	17	3
		4 dpf	20	20	0
Homozygous	aISV	2 dpf	15	20	5
		3 dpf	11	16	5
		4 dpf	5	11	6
	vISV	2 dpf	7	17	10
		3 dpf	8	19	11
		4 dpf	13	23	10

12. Line 362, Fig 6g,h: Text states that both ISV types have “significantly enlarged lumens at 2, 3, and 4 dpf compared to wild-type and *krit1*^{+/-} EC-containing vessels.” This is untrue according to statistical analysis.

After analysing more vessels from transplantation experiments (see table below), we have corrected this statement to clarify that vessel diameters were larger in *krit1*^{-/-} aISVs at 2, 3 and 4 dpf and vISV at 3 and 4dpf, compared to wildtype controls (Supplementary Fig. 7b, lines 398-400).

Genotype	Vessel type	Developmental stage	No. of vessels (first submission)	No. of vessels (resubmitted)	No. of increased samples
Wild type	aISV	2 dpf	3	8	5
		3 dpf	6	9	3
		4 dpf	9	12	3
	vISV	2 dpf	5	6	1

		3 dpf	3	6	3
		4 dpf	5	10	5
Heterozygous	aISV	2 dpf	15	21	6
		3 dpf	7	11	4
		4 dpf	7	8	1
	vISV	2 dpf	6	14	7
		3 dpf	5	7	2
		4 dpf	9	9	0
Homozygous	aISV	2 dpf	7	11	4
		3 dpf	5	8	3
		4 dpf	4	6	2
	vISV	2 dpf	5	14	9
		3 dpf	5	13	8
		4 dpf	7	13	6

13. The DA and PCV also get smaller in diameter over early developmental time. Can you speculate on whether your findings are likely applicable to larger multicellular tubes?

To address this question, we have imaged endothelial cells in both DA and PCV. We also found the existence of circumferential actin bundles as well as actomyosin asters, suggesting that larger vessels generate different actin structures, including circumferential actin, for vessel remodelling (Supplementary Fig. 8, lines 459 - 465).

14. Reference 15 is now published, please update.

Thank you for pointing out that this paper is now published. Its reference is now updated.

Reviewer #2 (Remarks to the Author):

This manuscript by Chen et al. addresses the mechanisms that lead to zebrafish intersegmental vessels remodeling during development. Whether and how acto-myosin contraction allows vessel constriction during vessel remodeling is the driving question. A recent paper in *Angiogenesis* by Yin et al. 2024, has investigated the role of acto-myosin contraction at cell-cell junctions for lumen formation in zebrafish without addressing the question of vessel caliber. Therefore, the questions here have yet not been addressed directly.

However, the general claim that circumferential actomyosin bundles and not the other actin organizations drive endothelial cell deformations is not enough demonstrated. Moreover, the contribution of other described phenomena such as cell exchange and cell division to vessel constriction is not well taken into account.

Major comments:

1- First, it is difficult to follow the terminology describing the different events. The exact meaning of terms like cell rearrangement, cell exchange, cell shape deformation, cell size and their relationship is not clear and makes figure 1 difficult to follow. Terms should be more clearly explained and maintained consistent between methods and the main results section. The results discuss the *tissue* strain while the methods describe the calculations for a *vessel* strain which can be confusing.

We have extensively rewritten the manuscript to improve the clarity of the different cellular behaviours in relation to changes in vessel morphometrics (diameter and length). We have also separated the former Figure 1 into two figures:

- The new Figure 1 focuses of vessel morphometrics and cell exchanges and division. We have also included an illustration to explain what is meant by cell exchange.
- The new Figure 2 is dedicated to describing changes in cell shape. We have included illustrations to summarise changes in cell area and data from strain analysis.

While rewriting the manuscript, we have paid attention to being consistent in our choice of terms. Accordingly, tissue strain has now been changed to vessel strain.

Current term	Previous term
Vessel strain	Tissue strain
Cell strain	Cell strain
Cell number strain	Effective cell strain due to number/ E.number
Radial axis	Radial axis/minor axis
Axial axis	Vertical axis/major axis

2- The method description needs to be much more clear and a schematic should be added to figure 1 to explain how the geometry of the cell is measured with an ellipse depicted with the major and minor axis which then leads to the various calculations described. Cv and Cr should be more clearly described as the length of the radial and vertical axes of the ECs assuming an ellipse.

We have now included a schematic to explain how the radial and axial axes of the cell (Fig. 2a). In the revised manuscript, we use the term “radial axis” to refer to the “minor axis”, and “axial axis” to refer to the “major/vertical axis”. We have also included a more detailed explanation on Methods in the section ‘In vivo cell shape analysis’. Cr is cell strain at the radial axis and Cv is cell strain at the vertical axis. In lines 674-680, we explained that cell at radial axis is cell width (w) and at axial axis is cell length (l). For a quantity X (cell width w or cell length l), strain (ϵ) between two time points (t) was calculated as:

$$\epsilon_X(t_1 \rightarrow t_2) = \frac{X_2 - X_1}{X_1}$$

For example, cell strain at radial axis at 2-3dpf, would be calculated as:

$$\epsilon_w(t_{2dpf} \rightarrow t_{3dpf}) = \frac{w_{3dpf} - w_{2dpf}}{w_{2dpf}}$$

to indicate the change of cell width at the circumference between two timepoints.

3- In figure 2b, it is quite difficult to follow the variation in diameter of the vessels at the front and back sides. A curve representing actin intensity and vessel diameter over time as in figure 4c would help. The respective contribution of actin rings and cell rearrangement (with new cells entering the zone) is not clear.

The intention of this figure (new Fig. 3b) is to illustrate the dynamics of cell rearrangement within the circumference of the vessel at two regions. Region 1 (demarcated by serrate pink line) is a unicellular segment of the vessel while region 2 (demarcated by serrated blue line) is a multicellular segment of the same vessel. We showed the front and back sides of the vessel to showcase the junction organization so to highlight the differences in junctions at the front and back sides. We appreciate that this may be confusing and have added a maximum projection of the vessel in this figure. We have also rearranged the figure to highlight the dynamics of the two cells at the cross-sections of the 2 regions. We hope that these additional changes will help the reviewer understand the images better.

As the purpose of this figure is not to correlate the intensity of actin and vessel diameter, we do not see that it is fitting to place a curve representing actin intensity and vessel diameter over time here. We would like to refer the reviewer to new Supplementary Figure 5b for 16 curves showing the relationship between circumferential actin (and myosin II) intensity and

vessel diameter. These curves consistently show that high circumferential actin intensity is associated with decreased vessel diameter and vice versa.

We are unsure what is meant by “actin rings”. We assume that the reviewer is referring to the cortical actin stripes that we investigated further by higher resolution imaging (from new Figure 4 onwards) and classified them as circumferential actin bundles. Although these actin structures are circumferential, they are not rings because they do not surround the entire circumference of the vessel. In this study, we did not examine the relationship between its formation and cell rearrangement but focused on the role of circumferential actin in driving cell contraction and consequently, vessel constriction.

4- The following part of the manuscript describes the role of actin remodeling on vessel constriction. However, this relies on a manual annotation of three types of actin which is a highly biased method. For example in fig3c, d, fig 5, fig6, the manual annotation of the actin fibers does not seem very close to the reality. Actin segmentation and categorization should be automated to make it more reliable.

We agree that there may be a degree of bias in characterizing the actin organization subjectively, and three members of the lab have independently attempted to automate the analysis of actin structures detected in endothelial cells *in vivo*. However, each person encountered the same problem. Due to a high spectrum of actin intensity, from very high junctional signal to weakly labelled cortical actin structures that cannot be detected by automated thresholding (e.g. Otsu method), the number and type of actin filaments generated by image processing and segmentation do not represent what is detectable by eye. Thus, although a lot of time was invested to develop an image analysis method to annotate actin organization objectively, it did not yield reliable result, and we concluded that the establishment of a robust and reliable automated method of actin organisation requires immense work and constitutes its own project. We therefore resorted to the analysis of actin organisation by eye, and the circumferential-to-longitudinal transition between 2 and 4 dpf was observed by two other researchers independently, supporting reproducibility.

After receiving reviewers’ comments, we have again tried to automate the detection of actin orientation and established an automated nematic-orientation workflow (lines 528 – 533, Supplementary Fig. 11, Supplementary Method 1). Actin orientations were binned into three classes (-1 to -0.33 = circumferential (C); -0.33 to $+0.33$ = mesh; $+0.33$ to 1 = longitudinal (L)) to generate frequency plots. While this analysis broadly supports a temporal transition in actin organisation, it generally reports fewer C and more (L) categories than our five-bin manual scheme. This discrepancy most likely reflects method-intrinsic factors: first, the uniform thresholding that removes junctional signal can also exclude adjacent thick circumferential bundles, reducing apparent C; second, referencing θ to a single overall vessel axis on curved or asymmetric segments biases near-circumferential orientations towards 0 while preserving streaks near $+1$, right-shifting medians into the L bin; third, per-vessel median summarisation compresses mixed distributions, so vessels that appear

mesh-dominant can cross the +0.33 boundary and be labelled as L; last, intermediate states are collapsed by design (manual C-M and M-L map to mesh and L, respectively), inflating those bins at the expense of C. Despite these differences, both approaches support a time-dependent transition in actin organisation during remodelling. Future work should prioritise segmentation that captures the full intensity dynamic range of cortical actin and calculation of orientation relative to the local centreline tangent rather than a global axis.

We have clarified our manual classification in lines 686-689 and included images of C-M and M-L actin organizations in Fig. 4a to illustrate how the two classifications were determined.

In Fig 3, the percentage of the various actin organizations is quantified. But then the spatiotemporal relationship between actin organization and vessel/cell shape changes is only described for circumferential actin. Are the other actin types also involved ? if not, this needs to be shown to claim that it is primarily circumferential actin that drives vessel constriction.

We have plotted the diameter of vessels exhibiting circumferential, mesh and longitudinal actin on the same time scale in Supplementary Fig. 4. These additional analyses show that in 12 of 12 movies, the formation of circumferential actin correlates with decrease in vessel diameter. In 8/12 cases, the dissipation of circumferential actin is followed by vessel widening. In contrast, the formation of mesh actin formation is correlated with vessel widening (5 of 6 movies). Longitudinal actin formation is associate with varying degree of vessel constriction. This is mentioned in lines 217 - 224. Based on these correlative analyses, circumferential actin appears to be the primary actin that drives vessel constriction.

To strengthen the causal role of circumferential actin in vessel constriction, we have performed additional experiments:

- i) Endothelial overexpression of Fascin 1, an actin bundling protein (Fig. 6a -d, lines 302 - 313). These new data show that Fascin 1-OE increased the number of vessels with circumferential actin formation and a corresponding augmented constriction of aISVs and vISVs.
- ii) Endothelial overexpression of Krit1 (Fig. 7b-d, lines 380 - 386). Elevated Krit1 expression also resulted in more vessels with circumferential actin and enhanced vessel constriction.
- iii) Mathematical simulations of F-actin, myosin and crosslinkers enclosed by a membrane revealed that the stress generated in the circumferential direction was larger than that in the longitudinal direction, and that the generated stress can deform the membrane (Fig. 5d-g, lines 254 - 284).

5- This brings up the numbers of technical compared to biological replicates which are not clear. Indeed, there is no clear reporting throughout the entire paper of independent trials/embryos relative to the vessels or cells/vessel quantified. The “n” is reported

above several bar plots but very unclear what this n includes. Moreover, there is no appropriate statistical reporting section in the methods where this should also be elaborated upon. On first glance, since source data files are missing, the p values look very low for the distribution that can be seen in the bar plots.

We have now written the number of ISVs/cells, embryos and independent experiments, and statistical analysis used in all figure legends. Statistical analysis is added in methods section in lines 762-768. Source data is now provided in the resubmission.

6- Lines 308-309 conclude that circumferential actin is essential for vessel constriction but the OE-vessels still constrict and the difference between control and OE is 17 versus 14% which seems hardly enough to support this conclusion.

We agree with the reviewer that the data from wasb-OE experiment weakly supports the role of circumferential actin in vessel constriction and have moved these data to Supplementary Fig. 6. We have now obtained new data (Fascin1-OE, Krit1-OE and mathematical simulation described above) that provide stronger evidence for the role of circumferential actin in vessel constriction.

7- One interesting point raised is the contribution of flow to vessel dilation in Krit1-/- embryos. This still open question in the field needs more investigation on how no flow exacerbates CCM phenotype, and in what ways - actin/cell shape/proliferation/migration. Therefore, the effect of flow on the Krit phenotype should be more thoroughly explored and quantified. A movie of krit1-/- ISV with flow or no flow should be shown.

We agree with the reviewer's point that the question of how flow reduces the severity of Krit1 LOF phenotype is an important but unresolved question. While we agree that the analyses of actin organization, cell shape, proliferation and migratory behaviours of *krit1*^{-/-} ECs in the presence and absence of blood flow would contribute to the understanding of the relationship between blood flow and CCM formation, it is beyond the scope of this study for two reasons. The first is that our primary goal is to understand the cellular mechanism of blood vessel remodelling, not the effects of flow on *krit1*^{-/-} ECs. The second is that the duration to complete experiments required to fully understand how flow regulates actin, cell shape, proliferation and migration in *krit1*^{-/-} ECs far exceeds the time allocated for the revision of the manuscript. We estimate that this will take 9 to 12 months given that these experiments require cell transplantations (a technically difficult experiment that only yields about 5% zebrafish with grafts composed of *krit1*^{-/-} ECs) and the detailed nature of the image analyses. Further, if we only add a movie of *krit1*^{-/-} ISV with flow or no flow (which the reviewer requested), it does not add much value to the existing data.

We have consulted this matter with the editor, who has agreed that addressing this specific comment is out of the scope of the paper. Instead, we have added a statement “Notably, cell enlargement was more severe in flow-deficient *krit1*^{-/-} cells (from in-cross, Supplementary Fig. 7h) than in transplanted cells exposed to flow, suggesting that blood flow may partially mitigate the severity of Krit1 loss-of-function, a cellular mechanism that merits further investigation” in lines 489 - 492.

Minor comments:

8- Fig. 4d: which dpf is this comparison performed on?

Figures showed embryos at 2dpf. We have now specified this in Figure 5 legend.

For vISVs this difference between recoil velocity of circumferential and longitudinal actin is really not significant and the sample size is also small. Are the vISVs from different embryos or within one independent trial?

We have reanalysed and performed more experiments to increase sample size. The updated results now show a significant difference in recoil velocity between circumferential and longitudinal actin in vISVs (P = 0.0131).

We have now clarified the sample size and vessel type in the figure legend: aISV n=12/8/9 and vISV n=13/9/10 in C/M/L, from 27 embryos in 3 experiments.

9- Lines 302-305- claim that there is a reduction in circumferential actin in both aISVs and vISVs at 2/3/4dpf in wasb OE ECs. This seems to not correlate with the box plots (eg. 3 and 4dpf for aISVs show higher C-M in the wasb-OE).

We have re-written the results from the wasb-OE experiments and have also placed this set of data in Supplementary Fig. 6. In the revised manuscript, we have shifted the focus to increased mesh actin (lines 318 - 322): “Wasb-OE increased the frequency of ISVs exhibiting mesh actin, interestingly, at the expense of longitudinal rather than circumferential actin (Supplementary Fig. 6a-b). In both aISVs and vISVs, this shift was most apparent at later developmental stages (at 4dpf, M vs other AOs, aISV, Fisher’s test aISV, p=0.04; vISV, p=0.02).

10- Lines 313-331: the section discusses the effects on cell and vessel strains from Myosin II inhibition but does not discuss the effects on cell number changes which are identified as an important factor in Fig. 1. A quantification of ES/number of cells could help clarify this.

We thank the reviewer for this important suggestion. To address it, we performed additional Myl9bA2A3-OE experiments and quantified endothelial cell number together with vessel length in both aISVs and vISVs. These data allowed us to calculate cell number strain

(previously, E.number), which reflects changes in how many cells span the vessel axis, providing direct insight into cell addition by cell proliferation and rearrangement dynamics. The results are now included in Fig. 6h-i and Supplementary Fig. 6e-f and discussed in lines 343 – 349 and 359 - 363. Incorporating these analyses clarifies that myosin II inhibition alters the temporal coordination between cell rearrangement, cell contraction, and vessel constriction.

Reviewer #3 (Remarks to the Author):

Summary of the study: Using the zebrafish model, the authors intent to address how transitions in endothelial cell (EC) number and shape are coordinated to define vessel diameter during development remains. Their excellent data revealed that EC deformations, rearrangements and transient formation of self-seam junctions play key roles in regulating cell number and vessel diameter. By performing high-resolution analysis of actin cytoskeleton organization disclosed the generation of tension bearing, circumferential actomyosin bundles in the endothelial cortex that drive EC deformation and vessel constriction. In a vascular disease model of CCM, they showed that the loss of circumferential actin bundles in *krit1/ccm1*-deficient ECs causes cell enlargement and impaired vessel constriction that culminate in dilated vessels. Their strong data support that circumferential actomyosin-driven EC deformations in controlling vessel size and in the prevention of vascular malformations.

This is a very novel and well performed study with high quality data. I only have a minor issue regarding the myosin II activity in CCM disease model. They showed inhibition of myosin II activity leads to disrupted cell deformation and increased vessel diameter. Yet, CCM samples exhibited increased pospho-MLC2 thus increased myosin activity. How these can be reconciled?

We thank the reviewer for the positive evaluation of our work and for pointing out the disparity in myosin II activity in published works on *ccm1*-deficient ECs and our zebrafish model. While some studies report ROCK-driven hypercontractility, others showed reduced junctional myosin II activity. To elucidate the contractile state of *krit1*^{-/-} ECs during vessel remodelling, we examined the localization of actin and myosin II using *Tg(fli1:GAL4FF)^{ubs3}; Tg(UAS:EGFP-UCHD)^{ubs18}; Tg(6xUAS:myl9b-mCherry)^{rk32}* (Supplementary Figure 7g, lines 422 - 431). We found that the loss of Krit1 alters the distribution of actin and myosin II at junctions versus cortex, potentially affecting force transmission and EC transmission. The differences in myosin II activity in *in vitro* and *in vivo* studies, and the implications of our findings, are now discussed in lines 475 -495.

Reviewer #4 (Remarks to the Author):

We thank all the reviewers for their insightful and important comments, which improved our manuscript tremendously. Please find our point-by-point response below.

REVIEWERS' COMMENTS

Reviewer #1 (Remarks to the Author):

The authors have very nicely addressed all of my previous concerns regarding the original submission, as evidenced by numerous additional experiments presented in both the main text and in the supplement. They convincingly demonstrate a role for circumferential actomyosin in endothelial cell contraction and vessel narrowing and new data strengthen support for a role for the CCM protein, KRIT1, in this process.

We thank the reviewer for the positive feedback on our revised manuscript and are delighted that they are satisfied with our revision.

Reviewer #2 (Remarks to the Author):

First we would like to acknowledge the authors for their efforts in thoroughly revising the manuscript. The use of a consistent terminology over the manuscript helps its understanding like the new explanations on the quantified parameters.

Additional data have been acquired for many of the figures improving the robustness of the statistical analyses and numbers of studied vessels, embryos and experiments have been specified as requested.

New experiments with overexpression of Krit and Fascin nicely complement the data. However, some parts still need attention:

We thank the reviewer for acknowledging our efforts during the revision process. Please find below our point-by-point response to additional concerns.

-We noticed that in Fig1c only one experiment has been performed to quantify EC number per vessel at 4 dpf which is surprising as many other quantifications have been done at this time point in other figures.

We thank the reviewer for highlighting this point. We agree that including an additional biological replicate at 4 dpf would improve the robustness of the quantification. We have performed an independent repeat experiment and updated the figure, figure legend (lines 1012 – 1013), statistical analyses and source data accordingly. The latest result with additional replicate is consistent with the previously observed increase in EC number between 2 and 4 dpf.

-In figure 2g, at 3-4 dpf, it is said that aEC shrinks while vEC shortens. Please explain what it means either directly in the figure or in its legend.

We thank the reviewer for pointing out the need for clarification. We have revised the legend of Fig. 2g to explicitly define the terminology used:

Shorten: cell length decreases at axial axis; Narrow: cell width decreases at radial axis; Shrink: reduces in both axial and radial axis.

This is revised in lines 1037- 1039.

-In figure 4c-f, it is not said whether these are aISV or vISV.

We clarified the vessel type in the figure legend. Lines 1063-1064.

-In the paragraph line 298-313, it should be mentioned that whereas circumferential actin is more enriched at 2 dpf, diameter constriction is only statistically significant at 3 and 4 dpf.

Thank you for the suggestion. We have now added this clarification to the corresponding paragraph in the Results section "Notably, circumferential actin enrichment was highest at 2 dpf, whereas statistically significant diameter constriction was detected only at 3 and 4 dpf." Line 318-320.

-Line 377: please cite the original paper from Glading et al., JCB 2007.

We apologise for the incorrect citation and have now cited the original paper from Glading et al., JCB 2007.

- Paragraph 422-431 and sup fig 7h: a new Junction/Cortex Ratio is introduced in the quantifications but the term cortex is not clearly defined. Does this "cortex actin" actually corresponds to the total intracellular actin content or to only a zone and which one ? Concerning the linear fibers that are present in *krit1*^{-/-} ECs (Fig 7g), why are they not considered as circumferential actin fibers ? They also cross the cell from one junction to another. We are not necessarily asking for more experiments but for more clarity about what is seen here and its interpretation. Obviously, the wrong location in z of these contractile fibers is key for the *krit1*^{-/-} phenotype and could explain the paradox raised by reviewer 3 concerning the increased pMLC in these cells.

The "cortex actin" measured in these analyses corresponds to a defined intracellular zone, not total actin content. Specifically, 10 cortical actin regions per vessel were manually selected from regions within the cell interior where EGFP-UCHD signal was present (but not contiguous with junctional pixels). Junctional actin was sampled at cell-cell contacts, where EGFP-UCHD intensity is highest. We agree that the spatial position of contractile fibres is critical, particularly in light of in vitro studies showing that CCM1/CCM2 loss promotes ROCK1-dependent ventral stress fibres. However, as endothelial cells in zebrafish vessels are extremely thin (~1-2 μm) and z-resolution is limited, apical and basal membranes cannot be reliably distinguished in our images; thus, the cortical region integrates both apical and basal surfaces. This definition has now been clearly described in the Methods section in line 739-752.

In our in vivo data, the prominent linear actin fibres observed in *krit1*^{-/-} ECs (Supplementary Fig. 7g) are clearly distinct from the circumferential bundles seen in wild-type cells. They exhibit a diagonal-to-longitudinal orientation along the vessel axis rather than the perpendicular orientation characteristic of circumferential bundles. We now specified this in line 437-439 and added arrows and arrowheads to indicate junctional and cortical actin, respectively, in Supplementary Fig. 7g). Also, most of these longitudinal actin structures do not connect to cell-cell junctions. Together with the reduced junction/cortex actin ratio and altered myosin II distribution, these observations indicate a redistribution of contractile

structures away from junctional-circumferential actin scaffolds toward a more homogeneous pattern. This pattern is consistent with the notion that increased pMLC in *krit1*^{-/-} cells does not translate into effective circumferential contraction because contractility is mis-localized and reoriented.

-Line 484: in Lisowska et al (ref 29), it is shown that in 2D, ROCK1 promotes ventral actin stress fibers connected to focal adhesions and ECM and not “cortical stress fibers” as said here.

We thank the reviewer for catching this inaccuracy and we apologize for the inappropriate phrasing. The text has now been corrected to refer specifically to “ventral actin stress fibres,” in line 504.

-Line 491: Please refer to fig 7e when talking about transplanted cells exposed to flow, it will facilitate the comparison. In this paragraph, mosaicism should also be commented as it may also impact the acto-myosin organization of the mutant cell and may contribute to the differences between pure homozygous vessels and transplanted populations.

We thank the reviewer for this helpful suggestion. We now explicitly refer to Fig. 7e when describing transplanted *krit1*^{-/-} cells exposed to flow. We also acknowledge that mosaicism may influence the actomyosin organization of transplanted mutant cells by placing them in a wild-type mechanical environment, and this clarification has now been incorporated into the revised manuscript in line 517-520.

-Line 539: When talking about the effect of CCM mutant cells on neighbouring ones, Shapeti et al, Nat com 2024 should also be cited.

Thank you for pointing out the relevance of the work of Shapeti et al. on the influence of CCM mutant cells on neighbouring wildtype cells. We have now cited this work in the manuscript (line 566).

-Movies 8 and 18 don't work.

We apologise for this technical error (they played on our computer). We have remade .avi files from .tif files for Movies 8 and 18 and hope that the reviewers can now open and view these movies.

Reviewer #3 (Remarks to the Author):

The authors have carefully addressed my previous concerns on Rock/MLC activation in CCM pathogenesis.

We are pleased that we have addressed the reviewer's concerns.

Reviewer #4 (Remarks to the Author):
